# Fairly Recommending with Social Attributes: A Flexible and Controllable Optimization Approach

**Jinqiu Jin**[1,*] **Haoxuan Li**[2,*] **Fuli Feng**[1,†] **Sihao Ding**[1] **Peng Wu**[3] **Xiangnan He**[1]

[1]University of Science and Technology of China
[2]Peking University    [3]Beijing Technology and Business University
jjq20021015@mail.ustc.edu.cn, hxli@stu.pku.edu.cn, fulifeng93@gmail.com,
dsihao@mail.ustc.edu.cn, pengwu@btbu.edu.cn, xiangnanhe@gmail.com

## Abstract

Item-side group fairness (IGF) requires a recommendation model to treat different item groups similarly, and has a crucial impact on information diffusion, consumption activity, and market equilibrium. Previous IGF notions only focus on the *direct* utility of the item exposures, *i.e.,* the exposure numbers across different item groups. Nevertheless, the item exposures also facilitate utility gained from the neighboring users via social influence, called *social* utility, such as information sharing on the social media. To fill this gap, this paper introduces two social attribute-aware IGF metrics, which require similar user social attributes on the exposed items across the different item groups. In light of the trade-off between the direct utility and social utility, we formulate a new multi-objective optimization problem for training recommender models with flexible trade-off while ensuring controllable accuracy. To solve this problem, we develop a gradient-based optimization algorithm and theoretically show that the proposed algorithm can find Pareto optimal solutions with varying trade-off and guaranteed accuracy. Extensive experiments on two real-world datasets validate the effectiveness of our approach. Our codes are available at https://github.com/mitao-cat/nips23_social_igf.

## 1   Introduction

Developing fair recommendation algorithms is crucial to perform reliable information search and decision making, which prevents users' as well as platforms' interests from being sacrificed [1]. One particular aspect of recommendation fairness is item-side group fairness (IGF), which requires a recommendation model to treat different item groups similarly, and the items are grouped based on attributes such as category or brand [2, 3, 4]. Existing IGF frameworks primarily focus on the *direct* utility of item exposures, requiring a similar number of exposures for different item groups, and can be broadly categorized into two forms: Statistical Parity (SP) [5, 6] and Equal Opportunity (EO) [7].

Nevertheless, the existing IGF notions overlook the *social* utility of item exposures, as users with different social attributes may produce varying utilities due to the social influence among users. For example, a user's friends may view a recommended micro-video due to sharing activities or public record in the user's timeline. This can lead to unfairness problem in which some item groups benefit more than others though the existing IGF notions are satisfied. Figure 1a illustrates a toy example where item group A receives the same number of exposures as group B, thus satisfying the requirement of the previous SP, while items in group A are recommended to users with greater numbers of friends. This motivates us to consider both direct and social utility for IGF notions.

---

*Equal contribution.
†Corresponding author.

37th Conference on Neural Information Processing Systems (NeurIPS 2023).

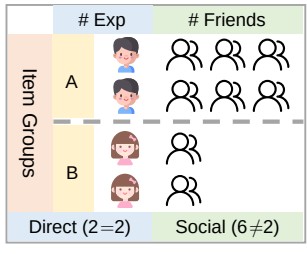 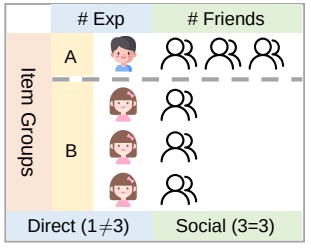 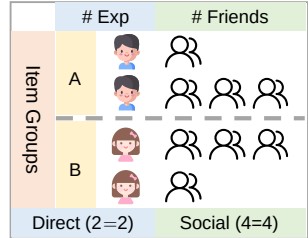

(a) Previous IGF notions      (b) Social attribute-aware notions      (c) Multi-objective optimization

Figure 1: A toy example with item groups A and B to illustrate direct and social utilities of the IGF notions, where "# Exp" is the number of exposed users across the item group, and "# Friends" is the number of friends of the exposed users. The multi-objective optimization considers both utilities.

In this paper, we first introduce two social attribute-aware IGF metrics, namely Neighborhood SP (NSP) and Neighborhood EO (NEO), which require that users exposed to different item groups have similar total social utility. As shown in Figure 1b, item groups A and B have the same total social utility, which is measured by the number of friends in this toy example. Compared to the previous SP and EO notions, NSP and NEO require similarity in the neighbors of the exposed users, rather than simply requiring the similar number of exposed users across the item groups. However, optimizing only the social attribute-aware IGF metrics may result in varying direct utilities across different item groups, *e.g.,* item groups A and B in Figure 1b have the different number of exposed users. Therefore, it is reasonable to consider both direct and social utility when developing fair recommender models.

To this end, we next formulate a multi-objective optimization problem for training the recommender models with consideration of both direct and social utility, and further require flexibility and controllability of the trade-offs. Specifically, motivated by the previous study [8], we incorporate pre-defined preference regions with varying trade-offs between the direct and social utility into the multi-objective optimization problem, enabling the flexibility to choose the desired trade-offs in practice. Meanwhile, we also incorporate a customizable accuracy constraint for the controllability of the optimization, which leads to a guaranteed recommendation accuracy while satisfying the IGF notions.

We further propose a **So**cial attribute-aware **F**lexible fair recommendation training algorithm with controllable **A**ccuracy (SoFA), for solving the above multi-objective optimization problem. Given a pre-defined preference region, the proposed gradient-based algorithm updates the model parameters within this region to achieve the desired trade-off. On the other hand, when the accuracy loss exceeds a given threshold, the model parameters update direction is changed to achieve the minimum accuracy guarantee. We also theoretically show that SoFA can find Pareto optimal solutions with varying trade-off, meaning the direct and social utility of the trained model is not dominated by any other solutions. Extensive experiments are conducted on two real-world datasets with user social attributes, and the experimental results validate the effectiveness of the proposed training algorithm.

The main contributions of this paper are summarized as follows:

- We propose two social attribute-aware IGF metrics, named NSP and NEO, to study the item exposure utility gained from the user social network.
- We formalize a multi-objective optimization problem to achieve flexible trade-off between the direct utility and social utility with controllable accuracy sacrifice of the recommendation.
- We further propose a gradient-based optimization algorithm to solve the above problem, called SoFA, and theoretically show that SoFA can find Pareto optimal solutions with varying trade-off.
- We conduct extensive experiments on two real-world datasets, revealing that methods ignoring social influence would lead to unfair exposure, and validating the effectiveness of our proposal.

## 2 Preliminaries

In this section, we present the background about personalized recommendation, item-side group fairness including SP and EO notions, and multi-objective optimization.

**Personalized Recommendation.** Personalized recommendation aims to select a small subset of items to each user $u$ based on historical data, with users denoted as $\mathcal{U} = \{u\}$ and items denoted as

$\mathcal{I} = \{i\}$. In our study, we focus on a common recommendation scenario known as collaborative filtering (CF) with implicit feedback, where the historical data contains the items that each user has interacted with [9, 10, 11, 12]. Bayesian Personalized Ranking (BPR) [9] is one of the representitive CF algorithm modeling the interaction probabilities, by minimizing the pairwise ranking loss

$$\min_{\boldsymbol{\theta}} \mathcal{L}_{\mathrm{BPR}}(\boldsymbol{\theta}) = - \sum_{(u,i,j)\in\mathcal{D}} \ln \sigma \left( \hat{y}_{u,i} - \hat{y}_{u,j} \right) + \frac{\lambda_{\boldsymbol{\theta}}}{2} \|\boldsymbol{\theta}\|_{\mathrm{F}}^2, \tag{1}$$

where $\mathcal{D}$ denotes the historical data for training, including user $u$, interacted item $i$, and no interacted item $j$. $\sigma(\cdot)$ is the Sigmoid function, $\boldsymbol{\theta}$ represents the model parameters, and $\lambda_{\boldsymbol{\theta}}$ is the regularization hyper-parameter to prevent overfitting. The predicted user preference scores $\hat{y}_{u,i}$ and $\hat{y}_{u,j}$ for positive item $i$ and negative item $j$ can be computed by using the widely-adopted matrix factorization [13].

**Item-Side Group Fairness (IGF).** The IGF requires fairness of recommendation holds across a set of item groups $\mathcal{G} = \{g_1, g_2, \ldots, g_A\}$, where each item $i \in \mathcal{I}$ belongs to one or more groups. The group of an item is correspond to attributes such as genre, brand, or other item characteristics, and denote $G_{g_a}(i)$ as the indicator function of whether item $i$ belongs to group $g_a$, *i.e.,* $G_{g_a}(i) = 1$ if $i \in g_a$, otherwise $G_{g_a}(i) = 0$. IGF notions aim to ensure fairness across different groups without over-recommending or under-recommending any specific group [14]. The existing IGF notions mainly focus on the exposure levels across different groups, and can be classified into positive rate-based metrics and confusion matrix-based metrics [1]. One particular positive rate-based metric is *Statistical Parity* (SP) [5, 6], which aims to ensure that each group has an equal likelihood of being recommended and requires the following quantities to be similar across different item groups

$$\mathbb{P}\left(R \mid g = g_a\right) = \frac{\sum_{u\in\mathcal{U}} \sum_{i\in\mathcal{I}} G_{g_a}(i) \cdot \hat{Y}(u,i)}{\sum_{u\in\mathcal{U}} \sum_{i\in\mathcal{I}} G_{g_a}(i)}, \quad \text{for all} \quad a = 1, \cdots, A, \tag{2}$$

where $\hat{Y}(u,i) = \sigma(\hat{y}_{u,i})$ estimates the interaction probability for user $u$ to item $i$, and $\sum_{i\in\mathcal{I}} G_{g_a}(i)$ is the total number of items belonging to group $g_a$. Instead, confusion matrix-based metrics further consider the ground truth labels. For instance, *Equal Opportunity* (EO) [7] aims to ensure the same true positive rate across different item groups

$$\mathbb{P}\left(R \mid g = g_a, y = 1\right) = \frac{\sum_{u\in\mathcal{U}} \sum_{i\in\mathcal{I}} G_{g_a}(i) \cdot Y(u,i) \cdot \hat{Y}(u,i)}{\sum_{u\in\mathcal{U}} \sum_{i\in\mathcal{I}} G_{g_a}(i) \cdot Y(u,i)}, \quad \text{for all} \quad a = 1, \cdots, A, \tag{3}$$

where $Y(u,i)$ is the ground truth label for user $u$ and item $i$, and $\sum_{i\in\mathcal{I}} G_{g_a}(i) Y(u,i)$ computes the total number of items that user $u$ has interacted with in group $g_a$. To keep the same value scale for different metrics, we evaluate the IGF of a recommendation model by computing the relative standard deviation of the group probabilities

$$\begin{aligned} \mathrm{SP} &= \mathrm{rsd}\left(\mathbb{P}\left(R \mid g = g_1\right), \cdots, \mathbb{P}\left(R \mid g = g_A\right)\right), \\ \mathrm{EO} &= \mathrm{rsd}\left(\mathbb{P}\left(R \mid g = g_1, y = 1\right), \cdots, \mathbb{P}\left(R \mid g = g_A, y = 1\right)\right), \end{aligned} \tag{4}$$

where $\mathrm{rsd}(\cdot) = \mathrm{std}(\cdot)/\mathrm{mean}(\cdot)$ calculates the relative standard deviation. The lower the SP and EO values, the fairer the recommender system satisfies the IGF fairness notions.

**Multi-Objective Optimization.** In many real-world tasks, there might be multiple optimization objectives rather than single one, while these objectives may collaborate or conflict with each other [15, 16]. Multi-objective optimization [17] aims to find a set of solutions that effectively balance the trade-off among these objectives, and it has been widely used in areas such as reinforcement learning [18] and E-commerce [19]. Typically, a multi-objective optimization problem is

$$\min_{\boldsymbol{\theta}} \mathcal{L}(\boldsymbol{\theta}) = \left(\mathcal{L}_1(\boldsymbol{\theta}), \mathcal{L}_2(\boldsymbol{\theta}), \cdots, \mathcal{L}_M(\boldsymbol{\theta})\right)^{\mathrm{T}}, \tag{5}$$

where $(\mathcal{L}_1(\boldsymbol{\theta}), \cdots, \mathcal{L}_M(\boldsymbol{\theta}))$ states the multiple objectives. To solve the multi-objective optimization problem, several population-based and evolutionary algorithm-based methods [20, 21, 22] have been proposed. Nevertheless, they are not efficient in handling large-scale datasets in recommendation. Alternatively, multi-objective gradient descent approches [23, 24] address this problem by leveraging Karush-Kuhn-Tucker (KKT) conditions [25] to find a gradient direction that reduces all objective values simultaneously. For example, gradient-based steepest descent methods [23] are proposed to seek a descent direction $\boldsymbol{d}_t$ by solving the following optimization problem

$$(\boldsymbol{d}_t, \alpha_t) = \arg\min_{\boldsymbol{d}, \alpha \in R} \alpha + \frac{1}{2} \|\boldsymbol{d}\|^2, \ s.t. \ \nabla\mathcal{L}_i(\boldsymbol{\theta})^T \boldsymbol{d} \leq \alpha, \ i = 1, \cdots, M, \tag{6}$$

then updates the model parameters using gradient descent $\boldsymbol{\theta}_{t+1} = \boldsymbol{\theta}_t + \eta \boldsymbol{d}_t$ with step size $\eta$. Theoretically, the solutions of the descent direction guarantee that all objective values will decrease, thus achieving the Pareto optimality [26]. To obtain these descent direction solutions, one can employ the widely-adopted multiple gradient descent algorithm (MGDA) [24].

## 3 Social Attributes-Aware IGF

Existing IGF notions such as SP and EO define the utility of recommending an item to a user only depends on the exposure and interaction numbers. For example, if item $i_1$ and item $i_2$ are recommended to user $u_1$ and user $u_2$ respectively, then both $i_1$ and $i_2$ have the same utility when $\hat{Y}(u_1, i_1) = \hat{Y}(u_2, i_2)$. However, they overlook the social attributes of users. Considering that the user social network plays a crucial role in click or conversion behaviors in real-world recommendation scenarios [27], recommending to users with distinct social attributes may lead to diverse item utilities.

To bridge this gap, we propose social attribute-aware IGF metrics as the extension of previously widely used SP and EO notions. Denote the neighbors (such as friends or followers) of user $u$ as $\mathcal{N}_u$, for each user $v \in \mathcal{N}_u$, let $R_v(i) > 0$ be the utility of item $i$ gained from user $v$ via the social influence of user $u$, when the item $i$ is recommended to the user $u$. Therefore, for positive rate-based metrics and confusion matrix-based metrics, the following quantities are required to be similar

$$\text{NR}\,(g = g_a) = \frac{\sum_{u \in \mathcal{U}} \sum_{i \in \mathcal{I}} G_{g_a}(i) \cdot \hat{Y}(u, i) \sum_{v \in \mathcal{N}_u} R_v(i)}{\sum_{u \in \mathcal{U}} \sum_{i \in \mathcal{I}} G_{g_a}(i)}, \quad \text{for all} \quad a = 1, \cdots, A,$$

$$\text{NR}\,(g = g_a, y = 1) = \frac{\sum_{u \in \mathcal{U}} \sum_{i \in \mathcal{I}} G_{g_a}(i) \cdot Y(u, i) \cdot \hat{Y}(u, i) \sum_{v \in \mathcal{N}_u} R_v(i)}{\sum_{u \in \mathcal{U}} \sum_{i \in \mathcal{I}} G_{g_a}(i) \cdot Y(u, i)}, \quad \text{for all} \quad a = 1, \cdots, A,$$

$$(7)$$

where the sum of utilities from the social network $\sum_{v \in \mathcal{N}_u} R_v$ are multiplied by the interaction probability $\hat{Y}(u, i)$ to measure the total social utility. Similar to Eq. (4), we then compute the *Neighborhood SP* and *Neighborhood EO* by the relative standard deviations

$$\begin{aligned} \text{NSP} &= \text{rsd}\,(\text{NR}\,(g = g_1), \cdots, \text{NR}\,(g = g_A)), \\ \text{NEO} &= \text{rsd}\,(\text{NR}\,(g = g_1, y = 1), \cdots, \text{NR}\,(g = g_A, y = 1)). \end{aligned} \quad (8)$$

**Further Discussion.** For the existing and the proposed IGF notions, the former including SP and EO focus on the direct utility obtained through recommendation exposures to users, whereas the latter including SP and EO emphasize the social utility gained from the user social network. Since these metrics emphasize different aspects, optimizing one IGF notion alone does not guarantee the optimality of the other IGF notion. Figures 1a and 1b illustrate this with a toy example: if we solely optimize SP, it does not guarantee the achievement of the other IGF notion, *i.e.,* NSP, and vise versa. Since both IGF metrics are important, it is desirable to optimize both SP (EO) and NSP (NEO) simultaneously from the prospective of multi-objective optimization, as illustrated in Figure 1c.

## 4 Methodology

### 4.1 Multi-Objective Optimization Problem Formulation

The optimization objective for training a fair recommendation model under IGF notions is to minimize both SP (EO) and NSP (NEO), so as to obtain similar direct utility and similar social utility simultaneously. Building upon prior studies [8, 28], we formulate the fair recommendation training task as a multi-objective optimization problem, where each optimization objective corresponds to a pre-defined IGF metric. Without loss of generality, let the number of IGF metrics used as optimization objectives be $M$, then we aim to obtain Pareto optimal solutions among these IGF metrics.

**Pareto Dominance.** A solution $\boldsymbol{\theta}_1$ dominates another solution $\boldsymbol{\theta}_2$ if $\mathcal{L}_i(\boldsymbol{\theta}_1) \leq \mathcal{L}_i(\boldsymbol{\theta}_2), \forall i = 1, \cdots, M$ and $\mathcal{L}(\boldsymbol{\theta}_1) = (\mathcal{L}_1(\boldsymbol{\theta}_1), \cdots, \mathcal{L}_M(\boldsymbol{\theta}_1))^{\text{T}} \neq \mathcal{L}(\boldsymbol{\theta}_2) = (\mathcal{L}_1(\boldsymbol{\theta}_2), \cdots, \mathcal{L}_M(\boldsymbol{\theta}_2))^{\text{T}}$.

**Pareto Optimality.** A solution $\boldsymbol{\theta}$ is a Pareto optimal solution if $\boldsymbol{\theta}$ is not dominated by any other solutions. The set of the Pareto optimal solutions is called as Pareto optimal set.

It is now attractive to formulate the trade-off between different IGF metrics as a multi-objective optimization problem, then obtain a Pareto-optimal solution between these metrics. However, due to the well-known accuracy sacrifice for improving fairness [14], directly solving the above problem without considering recommendation accuracy may lead to a significant reduction in recommendation quality. To ensure controllable recommendation accuracy during the optimization process, we introduce an accuracy constraint that penalizes instances with the accuracy loss (*e.g.,* BPR loss) exceeding a pre-defined threshold. Formally, the multi-objective optimization problem is

$$\min_{\boldsymbol{\theta}} \mathcal{L}(\boldsymbol{\theta}) = (\mathcal{L}_1(\boldsymbol{\theta}), \mathcal{L}_2(\boldsymbol{\theta}), \cdots, \mathcal{L}_M(\boldsymbol{\theta}))^{\mathrm{T}}, \ s.t. \ \mathcal{L}_{\mathrm{BPR}}(\boldsymbol{\theta}) \leq \xi, \tag{9}$$

where $\xi$ is the pre-defined accuracy threshold, and $\xi$ should be set to exceed the loss of a recommendation model trained using only BPR loss to ensure the existence of feasible solutions.

Furthermore, in real-world recommendation scenarios, the two types of IGF metrics that consider direct and social effects, respectively, may require different proportions of trade-offs. For example, a recommender system designed for social media applications may place more emphasis on the social utility than the direct utility. Therefore, it is meaningful to find flexible Pareto-optimal solutions to the multi-objective optimization problem with varying trade-offs. Following the previous work [8], we decompose the optimization problem into $N$ subproblems using a set of pre-defined unit preference vectors $\boldsymbol{s}_1, \boldsymbol{s}_2, \ldots, \boldsymbol{s}_N \in R_+^M$, and define the $N$ preference regions $\Omega_1, \Omega_2, \ldots, \Omega_N$ as follows

$$\Omega_n = \left\{ \mathcal{L}(\boldsymbol{\theta}) \in R_+^M \mid \boldsymbol{s}_j^T \mathcal{L}(\boldsymbol{\theta}) \leq \boldsymbol{s}_n^T \mathcal{L}(\boldsymbol{\theta}), \ \forall j = 1, \cdots, N \right\}, \tag{10}$$

where $\mathcal{L}(\boldsymbol{\theta}) \in \Omega_n$ if and only if $\mathcal{L}(\boldsymbol{\theta})$ forms the smallest acute angle with $\boldsymbol{s}_n$, resulting in the largest inner product $\boldsymbol{s}_n^T \mathcal{L}(\boldsymbol{\theta})$ among all $\boldsymbol{s}_1^T \mathcal{L}(\boldsymbol{\theta}), \boldsymbol{s}_2^T \mathcal{L}(\boldsymbol{\theta}), \ldots, \boldsymbol{s}_N^T \mathcal{L}(\boldsymbol{\theta})$. To ensure that the solution $\mathcal{L}(\boldsymbol{\theta})$ obtained from the optimization phase falls within the preference region $\Omega_n$, we further impose a constraint that penalizes the distance of the obtained solution from this preference region

$$\min_{\boldsymbol{\theta}} \mathcal{L}(\boldsymbol{\theta}) = (\mathcal{L}_1(\boldsymbol{\theta}), \mathcal{L}_2(\boldsymbol{\theta}), \cdots, \mathcal{L}_M(\boldsymbol{\theta}))^{\mathrm{T}}$$
$$s.t. \ \mathcal{G}_j(\boldsymbol{\theta}) = (\boldsymbol{s}_j - \boldsymbol{s}_n)^T \mathcal{L}(\boldsymbol{\theta}) \leq 0, \ \forall j = 1, \cdots, N, \tag{11}$$
$$\mathcal{L}_{\mathrm{BPR}}(\boldsymbol{\theta}) \leq \xi.$$

By letting the preference vectors be uniformly distributed in $R_+^M$, we can obtain flexible Pareto optimal solutions with varying trade-offs. Each subproblem corresponds to a unique Pareto-optimal solution in a specific preference region, which can reflect different attention to the direct and social utilities. Finally, we can select the desired Pareto optimal solution for diverse recommendation scenarios to satisfy both the flexible trade-offs and the controllable prediction accuracy.

### 4.2 Gradient-Based Flexible and Controllable Fair Recommendation Training

**Finding the Initial Solution.** For more efficient finding a solution to the multi-objective problem that falls within the specified preference region, we first find a feasible initial solution that satisfies all the constraints of that preference region. Specifically, for a randomly generated initial solution $\boldsymbol{\theta}_r$, we define the set of indices that violate the constraints as $I_\epsilon(\boldsymbol{\theta}_r) = \{j = 1, \cdots, N \mid \mathcal{G}_j(\boldsymbol{\theta}_r) \geq -\epsilon\}$, where $\epsilon$ is a small value to deal with solutions near the boundary. Then we can compute a descending direction $\boldsymbol{d}_r$ that reduces all the values of $\{\mathcal{G}_j(\boldsymbol{\theta}_r) \mid j \in I_\epsilon(\boldsymbol{\theta}_r)\}$ by solving the optimization problem

$$(\boldsymbol{d}_r, \alpha_r) = \underset{\boldsymbol{d}, \alpha \in R}{\arg\min} \ \alpha + \frac{1}{2}\|\boldsymbol{d}\|^2, \ s.t. \ \nabla\mathcal{G}_j(\boldsymbol{\theta}_r)^T \boldsymbol{d} \leq \alpha, \ j \in I_\epsilon(\boldsymbol{\theta}_r). \tag{12}$$

The gradient-based update rule can be expressed as $\boldsymbol{\theta}_{r_{t+1}} = \boldsymbol{\theta}_{r_t} + \eta_r \boldsymbol{d}_{r_t}$, where $\eta_r$ denotes the step size, and will stop when a feasible solution is found or the maximum iteration number is reached.

**Solving the Subproblem.** Given the feasible initial solution $\boldsymbol{\theta}_0$, we next propose a gradient-based learning approach to solve the multi-objective optimization problem in Eq. (11). Specifically, we compute the update direction $\boldsymbol{d}_t$ from $\boldsymbol{\theta}_t$ to $\boldsymbol{\theta}_{t+1}$ by solving the following optimization problem

$$(\boldsymbol{d}_t, \alpha_t) = \underset{\boldsymbol{d}, \alpha \in R}{\arg\min} \ \alpha + \frac{1}{2}\|\boldsymbol{d}\|^2$$
$$s.t. \ \nabla\mathcal{L}_i(\boldsymbol{\theta}_t)^T \boldsymbol{d} \leq \alpha, \ i = 1, \cdots, M,$$
$$\nabla\mathcal{G}_j(\boldsymbol{\theta}_t)^T \boldsymbol{d} \leq \alpha, \ j \in I_\epsilon(\boldsymbol{\theta}_t), \tag{13}$$
$$\nabla\mathcal{L}_{\mathrm{BPR}}(\boldsymbol{\theta}_t)^T \boldsymbol{d} \leq \alpha, \ \text{if} \ \mathcal{L}_{\mathrm{BPR}}(\boldsymbol{\theta}_t) \geq \xi.$$

The Lagrange function of the above optimization problem is

$$\mathcal{L}\left(\boldsymbol{d}, \alpha, \alpha_i, \beta_j, \lambda\right) = \alpha + \frac{1}{2}\|\boldsymbol{d}\|^2 + \sum_{i=1}^{M} \alpha_i \left(\nabla_{\boldsymbol{\theta}_t}\mathcal{L}_i(\boldsymbol{\theta}_t)^T \boldsymbol{d} - \alpha\right)$$

$$+ \sum_{j \in I_\epsilon(\boldsymbol{\theta}_t)} \beta_j \left(\nabla_{\boldsymbol{\theta}_t}\mathcal{G}_j(\boldsymbol{\theta}_t)^T \boldsymbol{d} - \alpha\right) + \lambda \cdot \mathbb{I}(\mathcal{L}_{\mathrm{BPR}}(\boldsymbol{\theta}_t) \geq \xi)\left(\nabla_{\boldsymbol{\theta}_t}\mathcal{L}_{\mathrm{BPR}}(\boldsymbol{\theta}_t)^T \boldsymbol{d} - \alpha\right), \tag{14}$$

where $\alpha_i \geq 0, \beta_j \geq 0, \lambda \geq 0$ are the Lagrange multipliers. By requiring that the derivatives of $\mathcal{L}\left(\boldsymbol{d}, \alpha, \alpha_i, \beta_j, \lambda\right)$ with respect to both $\boldsymbol{d}$ and $\alpha$ be zero, we have the following equations

$$\boldsymbol{d} = -\sum_{i=1}^{M} \alpha_i \nabla_{\boldsymbol{\theta}_t}\mathcal{L}_i(\boldsymbol{\theta}_t) - \sum_{j \in I_\epsilon(\boldsymbol{\theta}_t)} \beta_j \nabla_{\boldsymbol{\theta}_t}\mathcal{G}_j(\boldsymbol{\theta}_t) - \lambda \cdot \mathbb{I}(\mathcal{L}_{\mathrm{BPR}}(\boldsymbol{\theta}_t) \geq \xi)\nabla_{\boldsymbol{\theta}_t}\mathcal{L}_{\mathrm{BPR}}(\boldsymbol{\theta}_t),$$

$$\sum_{i=1}^{M} \alpha_i + \sum_{j \in I_\epsilon(\boldsymbol{\theta}_t)} \beta_j + \lambda \cdot \mathbb{I}(\mathcal{L}_{\mathrm{BPR}}(\boldsymbol{\theta}_t) \geq \xi) = 1. \tag{15}$$

Next, we compute the minimal value of Eq. (14) by substituting the corresponding terms in Eq. (15), and obtain the dual problem of Eq. (13) as $\max_{\alpha_i,\beta_j,\lambda} \min_{\boldsymbol{d},\alpha} \mathcal{L}\left(\boldsymbol{d}, \alpha, \alpha_i, \beta_j, \lambda\right)$, equals to

$$\min_{\alpha_i,\beta_j,\lambda} \frac{1}{2}\left\|\sum_{i=1}^{M} \alpha_i \nabla_{\boldsymbol{\theta}_t}\mathcal{L}_i(\boldsymbol{\theta}_t) + \sum_{j \in I_\epsilon(\boldsymbol{\theta}_t)} \beta_j \nabla_{\boldsymbol{\theta}_t}\mathcal{G}_j(\boldsymbol{\theta}_t) + \lambda \cdot \mathbb{I}(\mathcal{L}_{\mathrm{BPR}}(\boldsymbol{\theta}_t) \geq \xi)\nabla_{\boldsymbol{\theta}_t}\mathcal{L}_{\mathrm{BPR}}(\boldsymbol{\theta}_t)\right\|^2,$$

$$s.t. \sum_{i=1}^{M} \alpha_i + \sum_{j \in I_\epsilon(\boldsymbol{\theta}_t)} \beta_j + \lambda \cdot \mathbb{I}(\mathcal{L}_{\mathrm{BPR}}(\boldsymbol{\theta}_t) \geq \xi) = 1, \tag{16}$$

which can be efficiently solved by the gradient-based optimization methods such as MGDA [24].

Denote the solutions of Eq. (13) and Eq. (16) as $(\boldsymbol{d}^*, \alpha^*)$ and $(\alpha_i^*, \beta_j^*, \lambda^*)$, respectively. To obtain the solution of the dual problem, according to the KKT conditions, we have

$$\alpha_i^* \left(\nabla_{\boldsymbol{\theta}_t}\mathcal{L}_i(\boldsymbol{\theta}_t)^T \boldsymbol{d}^* - \alpha^*\right) = 0, \ i = 1, \cdots, M,$$

$$\beta_j^* \left(\nabla_{\boldsymbol{\theta}_t}\mathcal{G}_j(\boldsymbol{\theta}_t)^T \boldsymbol{d}^* - \alpha^*\right) = 0, \ \forall j \in I_\epsilon(\boldsymbol{\theta}_t), \tag{17}$$

$$\lambda \cdot \mathbb{I}(\mathcal{L}_{\mathrm{BPR}}(\boldsymbol{\theta}_t) \geq \xi)\left(\nabla_{\boldsymbol{\theta}_t}\mathcal{L}_{\mathrm{BPR}}(\boldsymbol{\theta}_t)^T \boldsymbol{d}^* - \alpha^*\right) = 0, \ \text{if } \mathcal{L}_{\mathrm{BPR}}(\boldsymbol{\theta}_t) \geq \xi.$$

By adding all equations in Eq. (17) together, we have $\alpha^* = -\|\boldsymbol{d}^*\|^2$. In this way,

- If $\boldsymbol{\theta}_t$ is Pareto optimal, then no other solution in its neighborhood can achieve better objective values, and we obtain the solution $\boldsymbol{d}^* = \boldsymbol{0}$, indicating that no direction can simultaneously improve all objective values.

- If $\boldsymbol{\theta}_t$ is not Pareto optimal, we have the following conclusions from Eq. (13)

$$\nabla\mathcal{L}_i(\boldsymbol{\theta}_t)^T \boldsymbol{d}^* \leq \alpha^* \leq -\|\boldsymbol{d}^*\|^2 < 0, \ i = 1, \cdots, M,$$

$$\nabla\mathcal{G}_j(\boldsymbol{\theta}_t)^T \boldsymbol{d}^* \leq \alpha^* \leq -\|\boldsymbol{d}^*\|^2 < 0, \ j \in I_\epsilon(\boldsymbol{\theta}_t), \tag{18}$$

$$\nabla\mathcal{L}_{\mathrm{BPR}}(\boldsymbol{\theta}_t)^T \boldsymbol{d}^* \leq \alpha^* \leq -\|\boldsymbol{d}^*\|^2 < 0, \ \text{if } \mathcal{L}_{\mathrm{BPR}}(\boldsymbol{\theta}_t) \geq \xi,$$

therefore $\boldsymbol{d}^*$ will be a descent direction that at least decreases all the IGF losses simultaneously, as well as the recommendation accuracy loss when $\mathcal{L}_{\mathrm{BPR}}(\boldsymbol{\theta}_t) \geq \xi$.

Finally, the model parameters are updated as $\boldsymbol{\theta}_{t+1} = \boldsymbol{\theta}_t + \eta\boldsymbol{d}_t$, where $\eta$ denotes the step size. Remarkably, throughout the optimization procedure, the values of all IGF objectives and the distance between the obtained solution and the preference region decrease, indicating the Pareto optimality within the preference region. Moreover, to tackle the problem of high-dimensional parameter $\boldsymbol{\theta}$ in the multi-objective optimization, we propose to optimize its dual problem rather than directly solving the primal problem. As a result, the numbers of parameters $(\alpha_i, \beta_j)$ in the dual problem is significantly reduced to be same as the numbers of IGF objectives and constraints, respectively, which greatly increases the scalability of the proposed algorithm to the large-scale recommendation datasets. We summarize the whole gradient-based fair recommendation training process in Algorithm 1.

**Algorithm 1:** **So**cial-Aware **F**lexible Fair Recommendation with Controllable **A**ccuracy (SoFA)

**Input:** Boundary threshold $\epsilon$, accuracy threshold $\xi$, step size $\eta_r$ and $\eta$.

1   Set the fairness preference vectors $\{s_1, s_2, \cdots, s_N\}$;

2   **for** $n = 1$ *to* $N$ **do**

3      Randomly initialize the model parameters $\boldsymbol{\theta}_r^{(n)}$;

4      Find the nearest feasible solution $\boldsymbol{\theta}_0^{(n)}$ from $\boldsymbol{\theta}_r^{(n)}$ for region $\Omega_n$;

5      **for** $t = 0$ *to* $T$ **do**

6          **if** $\mathcal{L}_{\mathrm{BPR}}(\boldsymbol{\theta}_t^{(n)}) \leq \xi$ **then**

7              Obtain $\alpha_i$, $i = 1, \cdots, M$, and $\beta_j$, $j \in I_\epsilon(\boldsymbol{\theta}_t^{(n)})$ by solving the dual optimization;

8              Compute the update direction
$$\boldsymbol{d}_t^{(n)} = -\sum_{i=1}^M \alpha_i \nabla_{\boldsymbol{\theta}_t^{(n)}} \mathcal{L}_i(\boldsymbol{\theta}_t^{(n)}) - \sum_{j \in I_\epsilon(\boldsymbol{\theta}_t^{(n)})} \beta_j \nabla_{\boldsymbol{\theta}_t^{(n)}} \mathcal{G}_j(\boldsymbol{\theta}_t^{(n)});$$

9          **else**

10              Obtain $\alpha_i, i = 1, \cdots, M$, $\beta_j, j \in I_\epsilon(\boldsymbol{\theta}_t^{(n)})$, and $\lambda$ by solving the dual optimization;

11              Compute the update direction $\boldsymbol{d}_t^{(n)} =$
$$-\sum_{i=1}^M \alpha_i \nabla_{\boldsymbol{\theta}_t^{(n)}} \mathcal{L}_i(\boldsymbol{\theta}_t^{(n)}) - \sum_{j \in I_\epsilon(\boldsymbol{\theta}_t^{(n)})} \beta_j \nabla_{\boldsymbol{\theta}_t^{(n)}} \mathcal{G}_j(\boldsymbol{\theta}_t^{(n)}) - \lambda \nabla_{\boldsymbol{\theta}_t^{(n)}} \mathcal{L}_{\mathrm{BPR}}(\boldsymbol{\theta}_t^{(n)});$$

12          **end**

13          Update the model parameters $\boldsymbol{\theta}_{t+1}^{(n)} = \boldsymbol{\theta}_t^{(n)} + \eta \boldsymbol{d}_t^{(n)}$;

14      **end**

15   **end**

## 5   Experiments

In this section we aim to answer the following research questions:

- **RQ1:** How does our proposed method perform compared with the existing approaches?

- **RQ2:** Do the preference regions in our method lead to more flexible Pareto solutions? Does the accuracy constraint result in more accurate recommendations while achieving IGF?

- **RQ3:** How does varying the accuracy thresholds affect the performance of our method?

### 5.1   Expertiment Setups

**Datasets and Pre-processing.** The experiments are performed on two benchmark recommendation datasets that include user social attributes and item features:

- **KuaiRec** is a short-video recommendation dataset on the video-sharing platform Kuaishou[3], which contains interactions between users and videos. In our experiments, we select user clicks spanning 10 consecutive days as the outcome of interest [29] and apply 10-core filtering to eliminate users and items with minimal interactions, which results in a pre-processed dataset consisting of 7,010 users, 1,405 items, and 863,819 interactions. For each user, the number of followers of that user is treated as the user social attribute, and we define the social utility derived from the user social attribute as the total fans count of that user. As to videos, we use tags to categorize them into six item groups.

- **Epinions** is a dataset derived from a trust network, comprising users' rating scores for products. Following previous work [10], we only keep the interactions with ratings greater than 3 and also adopt the 10-core filtering setting. Subsequently, we obtain a dataset consisting of 11,710 users, 13,682 items, and 215,262 interactions. For each user, the dataset includes their trust relations with other users, while for each product, it specifies its category. We let user's social utility equal to the number of trustors, and classify the products into six groups based on their categories.

Throughout our experiment, we randomly split the interactions of each dataset into training set (60%), validation set (20%), and testing set (20%). We use the validation set for tuning hyper-parameters and the testing data for evaluation the prediction and fairness performance of the trained models.

---

[3] https://www.kuaishou.com/.

Table 1: Performance comparison using SP and NSP as IGF notions, where SoFA is implemented with five preference regions. The best and second best results are bolded and underlined, respectively.

| | KuaiRec | | | | Epinions | | | |
|---|---|---|---|---|---|---|---|---|
| | N@5↑ | SP↓ | NSP↓ | F1SP↓ $_{deg}$ | N@5↑ | SP↓ | NSP↓ | F1SP↓ $_{deg}$ |
| BPRMF | **0.2426** | 0.0966 | 0.1119 | 0.1037 $_{49.2°}$ | 0.0443 | 0.0252 | 0.0286 | 0.0268 $_{48.6°}$ |
| + SP Reg | 0.2389 | 0.0062 | 0.0168 | 0.0091 $_{69.7°}$ | 0.0450 | 0.0140 | 0.0196 | 0.0163 $_{54.5°}$ |
| + NSP Reg | 0.2279 | 0.0366 | 0.0142 | 0.0205 $_{21.2°}$ | 0.0378 | 0.0224 | 0.0188 | 0.0205 $_{40.0°}$ |
| + SP&NSP Reg | 0.2369 | 0.0090 | 0.0245 | 0.0132 $_{69.8°}$ | 0.0448 | 0.0154 | 0.0205 | 0.0176 $_{53.2°}$ |
| + SP Post | 0.2412 | 0.0388 | 0.0545 | 0.0454 $_{54.5°}$ | 0.0445 | 0.0141 | 0.0196 | 0.0164 $_{54.2°}$ |
| + NSP Post | 0.2348 | 0.0844 | 0.0311 | 0.0455 $_{20.3°}$ | 0.0398 | 0.0212 | 0.0185 | 0.0197 $_{41.2°}$ |
| + SP&NSP Post | 0.2405 | 0.0817 | 0.0562 | 0.0666 $_{34.5°}$ | 0.0443 | 0.0152 | 0.0207 | 0.0175 $_{53.7°}$ |
| MOOMTL | 0.2229 | 0.0069 | 0.0238 | 0.0107 $_{73.8°}$ | 0.0446 | 0.0138 | 0.0193 | 0.0161 $_{54.4°}$ |
| SoFA $_{region\ 0}$ | 0.2349 | 0.0296 | **0.0096** | 0.0145 $_{18.0°}$ | 0.0364 | 0.0909 | 0.0294 | 0.0445 $_{17.9°}$ |
| SoFA $_{region\ 1}$ | 0.2376 | 0.0179 | 0.0105 | 0.0133 $_{30.4°}$ | 0.0441 | 0.0326 | 0.0225 | 0.0266 $_{34.6°}$ |
| SoFA $_{region\ 2}$ | 0.2329 | 0.0103 | 0.0146 | 0.0121 $_{54.8°}$ | **0.0451** | 0.0153 | 0.0210 | 0.0177 $_{53.9°}$ |
| SoFA $_{region\ 3}$ | 0.2413 | 0.0074 | 0.0194 | 0.0107 $_{69.1°}$ | 0.0427 | 0.0118 | **0.0177** | **0.0142** $_{56.3°}$ |
| SoFA $_{region\ 4}$ | 0.2402 | **0.0046** | 0.0227 | **0.0077** $_{78.5°}$ | 0.0185 | **0.0095** | 0.0314 | 0.0146 $_{73.1°}$ |

Table 2: Performance comparison using EO and NEO as IGF notions, where SoFA is implemented with five preference regions. The best and second best results are bolded and underlined, respectively.

| | KuaiRec | | | | Epinions | | | |
|---|---|---|---|---|---|---|---|---|
| | N@5↑ | EO↓ | NEO↓ | F1EO↓ $_{deg}$ | N@5↑ | EO↓ | NEO↓ | F1EO↓ $_{deg}$ |
| BPRMF | **0.2426** | 0.0189 | 0.1031 | 0.0319 $_{79.6°}$ | 0.0443 | 0.1182 | 0.2059 | 0.1502 $_{60.1°}$ |
| + EO Reg | 0.2340 | 0.0044 | 0.1082 | 0.0085 $_{87.7°}$ | 0.0425 | 0.1126 | 0.1956 | 0.1429 $_{60.1°}$ |
| + NEO Reg | 0.2329 | 0.0147 | 0.0992 | 0.0256 $_{81.6°}$ | 0.0430 | 0.1289 | 0.1557 | 0.1410 $_{50.4°}$ |
| + EO&NEO Reg | 0.2346 | 0.0054 | 0.1035 | 0.0102 $_{87.0°}$ | 0.0432 | 0.1136 | 0.1465 | 0.1280 $_{52.2°}$ |
| + EO Post | 0.2371 | 0.0061 | 0.1082 | 0.0116 $_{86.8°}$ | 0.0437 | 0.1100 | 0.1850 | 0.1379 $_{59.3°}$ |
| + NEO Post | 0.2280 | 0.0143 | 0.0672 | 0.0236 $_{78.0°}$ | 0.0427 | 0.1274 | 0.1565 | 0.1405 $_{50.8°}$ |
| + EO&NEO Post | 0.2413 | 0.0133 | 0.1039 | 0.0235 $_{82.7°}$ | 0.0436 | 0.1131 | 0.1630 | 0.1335 $_{55.3°}$ |
| MOOMTL | 0.2332 | 0.0031 | 0.1003 | 0.0061 $_{88.2°}$ | 0.0444 | **0.1093** | 0.1449 | **0.1246** $_{53.0°}$ |
| SoFA $_{region\ 0}$ | 0.1635 | 0.1454 | 0.0557 | 0.0806 $_{21.0°}$ | 0.0224 | 0.3273 | 0.1494 | 0.2051 $_{24.5°}$ |
| SoFA $_{region\ 1}$ | 0.2077 | 0.1135 | 0.0558 | 0.0749 $_{26.2°}$ | 0.0353 | 0.1752 | **0.1394** | 0.1552 $_{38.5°}$ |
| SoFA $_{region\ 2}$ | 0.2254 | 0.0610 | **0.0528** | 0.0566 $_{40.9°}$ | **0.0448** | 0.1094 | 0.1473 | 0.1256 $_{53.4°}$ |
| SoFA $_{region\ 3}$ | 0.2241 | 0.0284 | 0.0713 | 0.0406 $_{68.3°}$ | 0.0443 | 0.1160 | 0.1604 | 0.1347 $_{54.1°}$ |
| SoFA $_{region\ 4}$ | 0.2352 | **0.0023** | 0.0685 | **0.0045** $_{88.1°}$ | 0.0447 | 0.1157 | 0.1574 | 0.1334 $_{53.7°}$ |

**Evaluation Protocols.** We evaluate the recommendation accuracy with NDCG@5 [30]. For evaluating IGF, we report both the previous fairness metrics and social attribute-aware metrics, where lower value indicates fairer performance. We study two settings with $(SP, NSP)$ and $(EO, NEO)$ as the IGF objectives, respectively. Under each setting, the overall fairness is assessed using the harmonic-mean of the two IGF metrics denoted as F1SP. In addtion, to quantify the trade-offs between the two IGF metrics, we calculate the angles formed by these metrics via $deg = \arctan(NSP/SP)$, where a lower angle degree signifies a greater emphasis on the social utility.

**Hyper-Parameter Settings.** For all compared methods, we use a pre-trained BPRMF [9] as the backbone with fixed optimal batch size and regularization coefficient, then fine-tune the model by using the baselines methods for achieving IGF notions. For the proposed SoFA, we set $N = 5$ as the number of preference vectors that equally divide the first quadrant, and tune the learning rate for finding the initial solution within $\{0.3lr, lr, 3lr\}$, where lr denotes the learning rate for the pre-trained BPRMF. This optimization step stops once a feasible solution is found. The learning rate for all methods is searched within $\{0.03lr, 0.1lr, 0.3lr, lr, 3lr, 10lr\}$, and the coefficients of IGF terms for regularization-based and post-processing baselines are searched within $\{0.1, 0.2, 0.5, 1, 2, 5\}$. The early stopping strategy is employed when the F1SP or F1EO value does not decrease over 5 epochs.

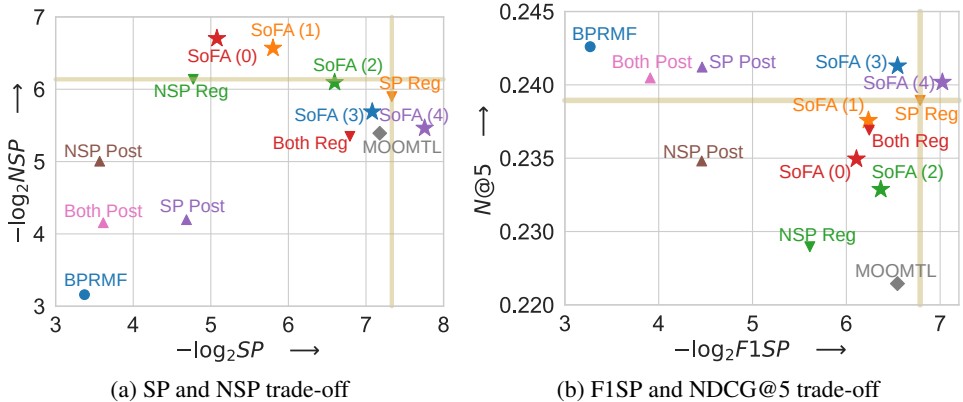

(a) SP and NSP trade-off

(b) F1SP and NDCG@5 trade-off

Figure 2: The trade-offs on KuaiRec between (a) IGF metrics, *i.e.,* SP and NSP, and (b) overall fairness and accuracy, *i.e.,* F1SP and NDCG, where yellow lines refer to the results when only optimizing one IGF metric, and scatters located on the upper right indicate better overall performance.

Table 3: Ablation studies of SoFA on the accuracy constraint, IGF objectives, and preference region.

| | KuaiRec | | | | Epinions | | | |
|---|---|---|---|---|---|---|---|---|
| | N@5↑ | SP↓ | NSP↓ | F1SP ↓ deg | N@5↑ | SP↓ | NSP↓ | F1SP ↓ deg |
| SoFA region 0 | 0.2349 | 0.0296 | 0.0096 | 0.0145 $_{18.0°}$ | 0.0364 | 0.0909 | 0.0294 | 0.0445 $_{17.9°}$ |
| w/o con | 0.2181 | 0.0415 | 0.0124 | 0.0191 $_{16.6°}$ | 0.0339 | 0.0814 | 0.0260 | 0.0395 $_{17.7°}$ |
| SoFA region 1 | 0.2376 | 0.0179 | 0.0105 | 0.0133 $_{30.4°}$ | 0.0441 | 0.0326 | 0.0225 | 0.0266 $_{34.6°}$ |
| w/o con | 0.2254 | 0.0211 | 0.0121 | 0.0154 $_{29.8°}$ | 0.0429 | 0.0371 | 0.0267 | 0.0311 $_{35.8°}$ |
| SoFA region 2 | 0.2329 | 0.0103 | 0.0146 | 0.0121 $_{54.8°}$ | **0.0451** | 0.0153 | 0.0210 | 0.0177 $_{53.9°}$ |
| w/o con | 0.2117 | 0.0085 | 0.0131 | 0.0103 $_{57.0°}$ | 0.0445 | 0.0163 | 0.0224 | 0.0189 $_{53.9°}$ |
| SoFA region 3 | 0.2413 | 0.0074 | 0.0194 | 0.0107 $_{69.1°}$ | 0.0427 | 0.0118 | 0.0177 | **0.0142** $_{56.3°}$ |
| w/o con | 0.2212 | 0.0077 | 0.0216 | 0.0113 $_{70.4°}$ | 0.0417 | 0.0131 | 0.0222 | 0.0165 $_{59.4°}$ |
| SoFA region 4 | 0.2402 | **0.0046** | 0.0227 | 0.0077 $_{78.5°}$ | 0.0185 | 0.0095 | 0.0314 | 0.0146 $_{73.1°}$ |
| w/o con | 0.2193 | 0.0069 | 0.0256 | 0.0109 $_{74.9°}$ | 0.0169 | **0.0093** | 0.0313 | 0.0144 $_{73.4°}$ |
| w/o SP | 0.2220 | 0.0206 | **0.0064** | 0.0098 $_{17.4°}$ | 0.0365 | 0.0216 | **0.0147** | 0.0175 $_{34.3°}$ |
| w/o NSP | **0.2415** | 0.0050 | 0.0173 | 0.0077 $_{74.0°}$ | 0.0447 | 0.0148 | 0.0200 | 0.0170 $_{53.5°}$ |
| w/o region | 0.2412 | 0.0046 | 0.0193 | **0.0074** $_{76.7°}$ | 0.0447 | 0.0141 | 0.0197 | 0.0164 $_{54.4°}$ |

## 5.2 Performance Comparison (RQ1)

We compare our proposed SoFA with the vanilla BPRMF [9] and several baseline methods to achieve recommendation fairness. These baselines include: (1) Regularization-based method [31], which incorporates IGF metrics as regularization terms into the BPR loss. (2) Post-processing method [32], which reconstructs the recommendation results while imposing fairness objectives as constraints. (3) Multi-objective optimization-based method [28], which optimizes the formulated multi-objective problem without the preference regions and recommendation accuracy constraint. We adopt three versions of regularization-based methods and post-processing methods, which considers only direct utility (SP or EO), only social utility (NSP or NEO), and both utilities, respectively.

Tables 1 and 2 show the overall performance of compared methods under the (SP, NSP) and (EO, NEO) settings, respectively, from which we have the following observations: (1) In all cases, SoFA outperforms other baselines in terms of accuracy and fairness, which validates its effectiveness in fairly recommending with accuracy guarantee. (2) SoFA can effectively find solutions in different preference regions, showing the ability to accommodate customized trade-off between the direct utility and social utility as the IGF metrics. Figure 2a visualizes the trade-off between SP and NSP, showing that SoFA is able to find pareto optimal solutions between the IGF metrics. Figure 2b suggests that F1SP and NDCG do not always exhibit a conflicting trade-off, meaning that enhancing fairness does not necessarily sacrifice recommendation accuracy in certain scenarios.

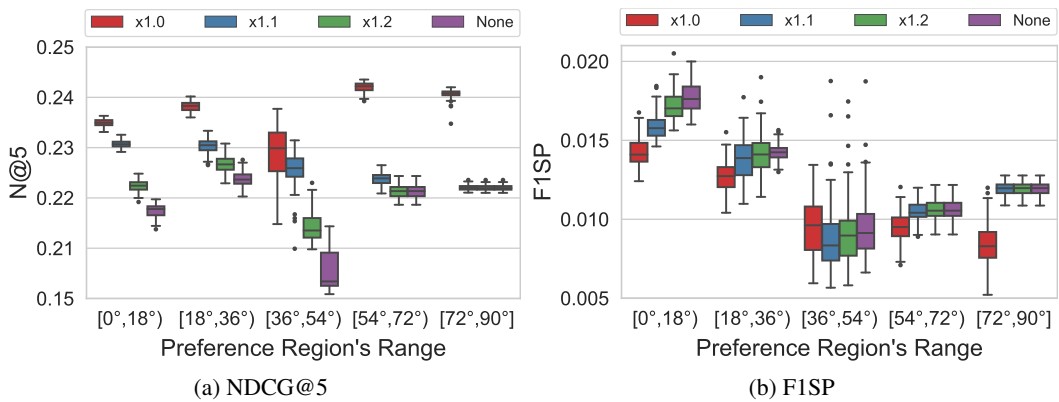

(a) NDCG@5          (b) F1SP

Figure 3: Effects of varying accuracy thresholds in the optimization on KuaiRec with 100 repeats.

## 5.3 In-depth Analysis

**Ablation Study (RQ2).** We conduct the following ablations to evaluate the effectiveness of specific designs in SoFA: (1) *w/o con*, which trains the model without the recommendation accuracy constraint. (2) *w/o region*, which trains the model without specifying the preference region. (3) *w/o SP*, which trains the model without the SP fairness loss, thus does not require the preference region specification. (4) *w/o NSP*, similar to *w/o SP* but trains the model without the NSP fairness loss. Table 3 shows the performance of SoFA and its ablated versions, we find that training the model without the accuracy constraint encounters significant drop in recommendation accuracy compared to SoFA, indicating the effectiveness of the proposed accuracy constraint. In addition, removing the specification of the preference region may not hurt the performance in terms of accuracy and fairness, because the preference region do not directly contribute to the recommendation accuracy and overall fairness. However, it fails to obtain a recommendation model with flexibility to trade-off these IGF metrics.

**Effects of Accuracy Requirement (RQ3).** To further investigate the effect of recommendation accuracy constraint, we evaluate SoFA with relaxed constraints by increasing the threshold $\xi$ in Eq. (9). In particular, we compare the default choice of the accuracy threshold with 1.1 times and 1.2 times of that value, as well as the optimization without the recommendation accuracy constraint. We conduct 100 runs for each optimization process on KuaiRec with varying preference regions. Figure 3 shows the accuracy and fairness performance, and we have observations as below: (1) NDCG has a clear decrease trend as the threshold increases, which further reveals that constraining the recommendation loss can control the accuracy of the obtained solution. (2) F1SP improves as the threshold decreases in most regions, which indicates that the accuracy constraint may also benefit the fairness. Therefore, we suggest to use the original BPRMF loss as the accuracy threshold in practice.

## 6 Conclusion

This study addresses a new problem in IGF motivated by the effect of user social influence on item utility. First, we propose two IGF notions, namely NSP and NEO, as extensions to the existing IGF notions that only considers the direct utility of item exposures. Next, we formulate a multi-objective optimization problem with pre-defined preference regions to ensure the flexibility of the trade-off between these IGF metrics. Then, we incorporate a recommendation accuracy constraint to control the accuracy sacrifice for satisfying the fairness requirements. We further propose an algorithm to solve the optimization problem and theoretically demonstrate its Pareto optimality. Extensive experiments on two real-world datasets validate the effectiveness of our method. One interesting future research direction is to generalize the user social influence studies in this paper to ranking-based IGF settings.

## Acknowledgement

This work is supported by the National Key Research and Development Program of China (No. 2022YFB3104701) and the National Natural Science Foundation of China (No. 12301370).

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
