# OpenReview forum: "Fairly Recommending with Social Attributes: A Flexible and Controllable Optimization Approach"
_NeurIPS.cc/2023/Conference — NeurIPS 2023 poster_

### Official Review · Reviewer_PYwN · 2023-07-01

**Soundness:** 3 good
**Presentation:** 3 good
**Contribution:** 3 good
**Rating:** 7
**Confidence:** 3

**Summary:**

Item-side group fairness (IGF) is crucial in recommendation tasks. Existing fairness notions focus on direct utility of item exposures but overlook the social utility, such as the influence of users' social attributes on exposed items. This paper introduces social attribute-aware IGF metrics and proposes a new IGF problem considering both direct and social utilities. To address the trade-off between these utilities, a new multi-objective optimization problem is formulated for training recommender models, allowing customizable trade-offs while maintaining controllable accuracy. A new optimization algorithm is developed with a theoretical guarantee of approaching restricted Pareto optimality. Experiments on real-world datasets confirm the effectiveness of this flexible and controllable approach.

**Strengths:**

Introducing social utility into group fairness if a reasonable assumption and this work proposed a viable solution with two new metrics. The experiments are thorough and detailed. This work is complete and sound.

**Weaknesses:**

1. Matrix factorization is not the mainstream choice for recommendation in practice. Tree model and deep model are the hot choice. Choosing them as backbone may further improve the impacts of this work.

**Questions:**

N/A

**Limitations:**

The authors have discussed the limitation.

---

> ### Author Rebuttal · Authors · 2023-08-10
>
> > **W1:** Matrix factorization is not the mainstream choice for recommendation in practice. Tree model and deep model are the hot choice. Choosing them as backbone may further improve the impacts of this work.
>
> **Response to W1:** Thanks for your helpful suggestion. **As suggested by the reviewer, we add extensive experiments by taking NeuMF and NGCF as backbones, respectively, to validate the effectiveness of our proposal** in below, to address your concerns of our work.
>
> | SP&NSP          | NDCG          | SP            | NSP           | F1SP          | deg  | EO&NEO          | NDCG          | EO            | NEO           | F1EO          | deg  |
> | --------------- | ------------- | ------------- | ------------- | ------------- | ---- | --------------- | ------------- | ------------- | ------------- | ------------- | ---- |
> | **NeuMF**       | 0.2220        | 0.2013        | 0.2464        | 0.2216        | 50.8 | **NeuMF**       | **0.2220**    | 0.0291        | 0.1006        | 0.0451        | 73.9 |
> | + SP Reg        | $\underline{0.2260}$ | 0.0157        | 0.0473        | 0.0235        | 71.7 | + EO Reg        | 0.2125        | 0.0034        | 0.1077        | 0.0066        | 88.2 |
> | + NSP Reg       | 0.2223        | 0.0229        | 0.0228        | 0.0228        | 44.9 | + NEO Reg       | 0.2191        | 0.0281        | $\underline{0.0363}$ | 0.0317        | 52.2 |
> | + SP&NSP Reg    | 0.2256        | 0.0165        | 0.0620        | 0.0261        | 75.1 | + EO&NEO Reg    | 0.2191        | 0.0053        | 0.1034        | 0.0100        | 87.1 |
> | + SP Post       | 0.2146        | 0.0151        | 0.0631        | 0.0243        | 76.6 | + EO Post       | 0.2183        | 0.0042        | 0.1073        | 0.0082        | 87.7 |
> | + NSP Post      | 0.2138        | 0.0686        | 0.0421        | 0.0522        | 31.5 | + NEO Post      | 0.2142        | 0.0185        | 0.0384        | 0.0250        | 64.3 |
> | + SP&NSP Post   | 0.2161        | 0.0187        | 0.0433        | 0.0262        | 66.6 | + EO&NEO Post   | 0.2168        | 0.0044        | 0.0721        | 0.0084        | 86.5 |
> | MOOMTL          | 0.1695        | 0.0177        | 0.0611        | 0.0274        | 73.9 | MOOMTL          | 0.2113        | $\underline{0.0034}$ | 0.1065        | $\underline{0.0065}$ | 88.2 |
> | Ours (region 0) | 0.1529        | 0.1064        | **0.0166**    | 0.0287        | 8.9  | Ours (region 0) | 0.1015        | 0.1130        | **0.0361**    | 0.0547        | 17.7 |
> | Ours (region 1) | 0.2237        | 0.0305        | $\underline{0.0171}$ | 0.0219        | 29.2 | Ours (region 1) | 0.1967        | 0.0830        | 0.0471        | 0.0601        | 29.6 |
> | Ours (region 2) | 0.2211        | 0.0540        | 0.0659        | 0.0594        | 50.7 | Ours (region 2) | 0.1917        | 0.1228        | 0.0926        | 0.1056        | 37.0 |
> | Ours (region 3) | 0.2136        | **0.0106**    | 0.0281        | **0.0153**    | 69.4 | Ours (region 3) | 0.2003        | 0.0744        | 0.1136        | 0.0899        | 56.8 |
> | Ours (region 4) | **0.2263**    | $\underline{0.0139}$ | 0.0484        | $\underline{0.0216}$ | 73.9 | Ours (region 4) | $\underline{0.2201}$ | **0.0017**    | 0.0941        | **0.0033**    | 89.0 |
>
> | SP&NSP          | NDCG          | SP            | NSP           | F1SP          | deg  | EO&NEO          | NDCG          | EO            | NEO           | F1EO          | deg  |
> | --------------- | ------------- | ------------- | ------------- | ------------- | ---- | --------------- | ------------- | ------------- | ------------- | ------------- | ---- |
> | **NGCF**        | 0.2389        | 0.1164        | 0.1356        | 0.1253        | 49.4 | **NGCF**        | **0.2389**    | 0.0210        | 0.1045        | 0.0349        | 78.7 |
> | + SP Reg        | 0.2372        | 0.0034        | 0.0202        | 0.0058        | 80.5 | + EO Reg        | 0.2351        | 0.0019        | 0.1110        | 0.0037        | 89.0 |
> | + NSP Reg       | 0.2318        | 0.0109        | 0.0114        | 0.0111        | 46.3 | + NEO Reg       | 0.2329        | 0.0092        | 0.0666        | 0.0161        | 82.2 |
> | + SP&NSP Reg    | 0.2392        | 0.0044        | 0.0185        | 0.0072        | 76.5 | + EO&NEO Reg    | 0.2338        | 0.0022        | 0.0763        | 0.0043        | 88.4 |
> | + SP Post       | 0.2378        | 0.0048        | 0.0253        | 0.0081        | 79.3 | + EO Post       | $\underline{0.2352}$ | 0.0027        | 0.1118        | 0.0054        | 88.6 |
> | + NSP Post      | 0.2353        | 0.0641        | 0.0369        | 0.0468        | 29.9 | + NEO Post      | 0.2349        | 0.0105        | 0.0706        | 0.0182        | 81.6 |
> | + SP&NSP Post   | 0.2376        | 0.0226        | 0.0221        | 0.0224        | 44.4 | + EO&NEO Post   | 0.2333        | 0.0021        | 0.0766        | 0.0041        | 88.4 |
> | MOOMTL          | 0.2256        | $\underline{0.0032}$ | $\underline{0.0092}$ | $\underline{0.0048}$ | 70.8 | MOOMTL          | 0.2106        | $\underline{0.0015}$ | 0.1060        | $\underline{0.0029}$ | 89.2 |
> | Ours (region 0) | 0.2135        | 0.0680        | 0.0201        | 0.0310        | 16.4 | Ours (region 0) | 0.1090        | 0.1782        | 0.1030        | 0.1305        | 30.0 |
> | Ours (region 1) | 0.2383        | 0.0214        | 0.0128        | 0.0160        | 30.9 | Ours (region 1) | 0.2065        | 0.0663        | **0.0338**    | 0.0448        | 27.0 |
> | Ours (region 2) | **0.2405**    | 0.0134        | 0.0120        | 0.0126        | 41.9 | Ours (region 2) | 0.1871        | 0.0353        | $\underline{0.0457}$ | 0.0398        | 52.3 |
> | Ours (region 3) | $\underline{0.2397}$ | 0.0152        | 0.0294        | 0.0201        | 62.6 | Ours (region 3) | 0.2200        | 0.0443        | 0.0694        | 0.0541        | 57.5 |
> | Ours (region 4) | 0.2155        | **0.0019**    | **0.0081**    | **0.0031**    | 76.6 | Ours (region 4) | 0.2347        | **0.0013**    | 0.1123        | **0.0026**    | 89.3 |

---

> > ### Comment · Reviewer_PYwN · 2023-08-19
> >
> > Thank you for your efforts on experiments. I've read all the threads and will keep my rating.

---

> > > ### Author Response · Authors · 2023-08-21
> > > **Thank you for your positive comments, and we kindly ask for your continued support of our work.**
> > >
> > > We thank the reviewer for the positive comments on our work as well as acknowledging the supplemental experiments. We kindly ask for your continued support of our work. Cheers!

---

### Official Review · Reviewer_ZQVW · 2023-07-04

**Soundness:** 2 fair
**Presentation:** 2 fair
**Contribution:** 3 good
**Rating:** 6
**Confidence:** 3

**Summary:**

This paper considered both direct and social utilities on the Item-side group fairness (IGF). The authors proposed a new flexible and controllable optimization approach to trade-off the utilities while ensuring accuracy. Firstly, they proposed two social attribute-aware IGF metrics, NSP and NEO by leveraging user’s social attributes. They then formulated a new multi-objective optimization (MOP) problem for training recommendation models with flexibility that incorporated a recommendation loss constraint into the MOP to maintain control over the optimization process on the two sets of IGF metrics. Additionally, they developed a new optimization algorithm to solve the MOP and achieve restricted Pareto optimality. Finally, they conducted extensive experiments on multiple real-world datasets to demonstrate the effectiveness of their proposed method. Please refer to the previous section for the strengths and weaknesses of this work.

**Strengths:**

1)	This paper focused on the Item-group fairness and considered the social utility by utilizing user’s social attributes, which has certain research value.
2)	This paper expanded the previous fairness metrics to proposed two new social attribute-aware IGF metrics that considered the the distribution of neighbors of users.
3)	This paper formulated a new MOP for training recommendation models to achieve controllable optimization and flexible optimization.
4)	This paper developed a new optimization algorithm and provide its theoretical guarantee of approaching restricted Pareto optimality to make reasoning process more rigorous and clear.
5)	This paper conducted extensive experiments over on real-world datasets to demonstrate the effectiveness of the proposed method.


**Weaknesses:**

1) In the introduction:
The introduction of relevant work is not detailed enough, mainly using legends to illustrate viewpoints and put forward new fairness indicators, but the illustration of legends is not very intuitive.

2) In the Experiments:
a. The datasets are not described.
b. Some other relevant fairness indicators should also be compared.
c. The analysis of the results of the model is not very detailed, and the proposed SP&NSP does not achieve a good result.
d. What do the four regions in Figure 2 refer to? Suggested description in conjunction with the datasets.


**Questions:**


1)	How is the number N of users aggregated in NSP and NEO determined?
2)	Whether the same inference can be achieved if other optimization objectives are used?
3)	Whether the user's social attributes themselves may contain certain sensitive attributes that cause unfairness in the group?


**Limitations:**

The authors argue that the limitation of the work in this paper is that it does not consider the ranking-based IGF metrics, they should careful thought needs to be given to what the two metrics proposed can do for their future work and how to build and solve the new MOP problem.

---

> ### Author Rebuttal · Authors · 2023-08-10
>
> Thank you very much for your thoughtful and comprehensive feedback on our paper. With your helpful comments, we have revised the **Introduction** and **Experiments** Sections with the aim of presenting the relevant work in a more systematic and clear manner, as well as adding missing experimental details.
>
> ### **Introduction**
>
> > W1: More detailed introduction of relevant work.
>
> **Response to W1:** We thank the reviewer for the useful comment, and **we'll include the following paragraph to the related work section in our revised version of paper, regarding the previous studies of fairness in recommendation.**
>
> "Recently, there have been growing attention on fairness issue in recommender systems [1,2]. The taxonomy of fairness notions in recommendation can be categorized from different aspects [3,4]. For example, as for different subjects, existing works usually eliminate unfairness in recommendation for user-side [5], item-side [13] and multi-side [6]; as for different granularities, the fairness  notions can be categorized into group fairness [5] and individual fairness [7]. In this work, we focus on item-side group fairness [8,9,10], which requires that different item groups should be treated similarly by the recommender system. Previous works on IGF usually pursues the Statistical Parity and Equal Opportunity of different item groups [11,12,13]. However, they overlook the social utility of item exposures. As users with varying social attributes may bring impact on their friends, each recommendation to different users may result in varying utilities. In this work, we introduce social attribute-aware IGF metrics, and further investigate on solving the trade-off between two sets of IGF metrics under controllable accuracy."
>
> ### **Experiments**
>
> > W2(a): Description of datasets.
>
> **Response to W2(a):** Due to the page limits, we **postpone the detailed dataset descriptions in Appendix B1 in supplementary material,** including their origination, processing procedures, statistics, grouping method and social attributes.
>
> > W2(b): Other relevant fairness indicators; More detailed result analysis.
>
> **Response to W2(b):** The reviewer raises an interesting concern. We add extensive experiments by taking NeuMF and NGCF as backbone model, respectively. Moreover, we set varying $R_v$ with different users, items, and user-item pairs, respectively, to validate the effectiveness of our method. Please kindly refer to the attachment PDF for more details.
>
> > W2(c): The proposed SP&NSP does not achieve a good result.
>
> **Response to W2(c):** We thank the reviewer for pointing out this issue. We believe that there is a misunderstanding. **In fact, "SP&NSP Reg (Post)" is one of our compared baselines,** which imposes both SP and NSP as regularization (post reconstruction) terms in the regularization-based (post-processing) method. In Table 1, all of our proposed method achieved best performance w.r.t. different fairness metrics.
>
> > W2(d): What do the four regions in Figure 2 refer to?
>
> **Response to W2(d):** As the reviewer pointed out, each subfigure of Figure 2 is divided into four regions by two reference lines, which represents the corresponding metric value of x/y-axis obtained by single-objective optimization methods. More specifically, in the 1st and 3rd subfigure, the hierarchical line refers to the -log2NSP value of NSP Reg baseline, and the vertical line refers to the -log2SP value of SP Reg baseline; in the 2nd and 4th subfigure, we take the NDCG and -log2F1SP value of SP Reg baseline for reference. As for the divided four regions, the solutions in upper right (lower left) region can achieve better (worse) performance than both two reference lines (i.e. Pareto dominance), and the solutions in upper right / upper left / lower right region are regarded as Pareto optimal solutions.
>
> ### **Questions**
>
> > Q1: Number of aggregated users in NSP/NEO
>
> **Response to Q1:** In our work, the number of aggregated users in NSP/NEO equals to $|\mathcal{N}_u|$, i.e., the number of the neighbors of user $u$.
>
> > Q2: Other optimization objectives
>
> **Response to Q2:** Setting other optimization objectives makes it difficult to theoretically achieve Pareto optimality, and the proposed method has strict theoretical guarantees. Please kindly refer to lines 185-190 in our manuscript.
>
> > Q3: user social attributes
>
> **Response to Q3:** This paper argues that previous studies on item-side group fairness (IGF) have lacked consideration of user-side utility that depends on a user's social attributes, rather than from the sensitive attributes prospective. And we will leave the problem of "the user's social attributes themselves may contain certain sensitive attributes that cause unfairness in the group" in our further work.
>
>
> ***
> **References**
>
> [1] Rectifying Unfairness in Recommendation Feedback Loop. SIGIR 2023.
>
> [2] Scoping Fairness Objectives and Identifying Fairness Metrics for Recommender Systems: The Practitioners’ Perspective. WWW 2023.
>
> [3] Fairness in Recommendation: A Survey. Arxiv 2022.
>
> [4] A Survey on the Fairness of Recommender Systems. TOIS 2023.
>
> [5] Improving Recommendation Fairness via Data Augmentation. WWW 2023.
>
> [6] Joint Multisided Exposure Fairness for Recommendation. SIGIR 2022.
>
> [7] FairGAN: GANs-based Fairness-aware Learning for Recommendations with Implicit Feedback. WWW 2022.
>
> [8] Fairness in Recommendation Ranking Through Pairwise Comparisons. KDD 2019.
>
> [9] Fairness-Aware Ranking in Search & Recommendation Systems With Application to LinkedIn Talent Search. KDD 2019.
>
> [10] Towards Long-term Fairness in Recommendation. WSDM 2021.
>
> [11] Towards a Fair Marketplace: Counterfactual Evaluation of the Trade-Off Between Relevance, Fairness & Satisfaction in Recommendation Systems. CIKM 2018.
>
> [12] Achieving Fairness via Post-processing in Web-Scale Recommender Systems. FACCT 2022.
>
> [13] ProFairRec: Provider Fairness-aware News Recommendation. SIGIR 2022.

---

> > ### Comment · Reviewer_ZQVW · 2023-08-21
> >
> > The authors have answered all my questions. I will keep my score unchanged.

---

> > > ### Author Response · Authors · 2023-08-21
> > > **Thank you for your positive comments, and we kindly ask for your continued support of our work.**
> > >
> > > Thank you for your positive comments on our work and we are glad that we were able to address all your questions. We kindly ask for your continued support of our work. Best wishes!

---

### Official Review · Reviewer_cJkK · 2023-07-04

**Soundness:** 2 fair
**Presentation:** 2 fair
**Contribution:** 2 fair
**Rating:** 5
**Confidence:** 4

**Summary:**

This paper aims to solve the Item-side Group Fairness (IGF) issue in Recommender Systems (RSs). The paper claims that existing IGF notions only focus on the direct utility of item exposures while ignore the influence from the social attributes of users. To tackle this issue, the paper proposes social attribute-aware IGF metrics and utilizes a multi-objective optimization method to enhance the fairness of a given recommendation model. Experiments on two public datasets have been conducted to prove the efficiency of the proposed method.

**Strengths:**

The strengths of this paper can be summarized into several aspects:
1. The paper addresses an important research area in machine learning, which is the fairness of recommendation systems. The inclusion of social attributes in the proposed IGF model adds an interesting dimension to this field.

2. The experimental results demonstrate the efficiency of the proposed method. The performance of the proposed method outperforms other baselines across all datasets, which highlights its effectiveness.

3. The utilization of a multi-objective optimization method to solve the problem presented in the paper is a convincing approach. This method ensures that the accuracy of the recommendation model is maintained while enhancing the fairness aspect.

**Weaknesses:**

The weaknesses of this paper can be summarized into several aspects:

Idea and technique
1. The effect of social attributes on IGF is not clear. The authors should provide more empirical studies using the given datasets and conduct further analysis to describe the relationship between social attributes and IGF. A toy example in Figure 1 is insufficient to help readers understand this issue. For a fairness problem, it is crucial to clearly describe the problem and demonstrate its importance.
2. While utilizing a multi-objective optimization method (e.g., Pareto optimization) is convincing, it is not novel enough. Similar methods have been widely used in recommender systems, like [1].

Presentation
1. I highly recommend that the authors give a proper name to their method, which would make it more convenient for referencing. (This is not a reason for my rejection).
2. Some mathematical notations are not clear in this paper, making it hard to follow. For example, $G_{g_a}(i)$ in line 82 should reflect its relationship to $u$. Besides, $R_v$ in line 117 should reflect its relationship to $u$ and $i$.

Experiment
1. Although BPRMF is a classical recommendation method, it is an outdated recommendation algorithm compared to deep-learning-based methods used in modern recommender systems. Therefore, mitigating fairness in BRPMF may have limited impact in today's systems. Exploring the fairness issue using popular deep-learning algorithms like NeuMF[2] and NGCF[3] may be more effective to evaluate the proposed method.
2. The value of $R_v$ is important for the proposed method. The authors should conduct experiments with different values of this hyperparameter to prove why they set the default value to 1.
3. In the ablation study, when optimizing the model with only SP (i.e., w/o SP), the model achieves the best NSP fairness. The paper should analyze the reason for this.

[1] Hao Q, Xu Q, Yang Z, et al. Pareto optimality for fairness-constrained collaborative filtering[C]//Proceedings of the 29th ACM International Conference on Multimedia. 2021: 5619-5627.
[2] He X, Liao L, Zhang H, et al. Neural collaborative filtering[C]//Proceedings of the 26th international conference on world wide web. 2017: 173-182.
[3] Wang X, He X, Wang M, et al. Neural graph collaborative filtering[C]//Proceedings of the 42nd international ACM SIGIR conference on Research and development in Information Retrieval. 2019: 165-174.

**Questions:**

1. Why this paper chooses BRPMF to conduct your experiments rather than some novel deep-learning-based methods like NeuMF and NGCF?
2. Why optimizing the model with only SP can achieve the best NSP fairness?
3. Why you didn’t conduct an experiment of the effect of the value of $R_v$?
------------------------------
Dear authors,
Thanks for your response. Some of my concerns have been addressed, and I would like to improve my rating.

---

> ### Author Rebuttal · Authors · 2023-08-09
>
> We sincerely thank you for the helpful suggestions. **As suggested by the reviewer, we have added extensive experiments to address all your concerns.** Below, we hope to address your concerns and questions to improve the clarity and quality of our paper.
>
> ### **Idea and technique**
> > **W1(a):** The effect of social attributes on IGF is not clear. The authors should provide more empirical studies using the given datasets and conduct further analysis to describe the relationship between social attributes and IGF. A toy example in Figure 1 is insufficient to help readers understand this issue. For a fairness problem, it is crucial to clearly describe the problem and demonstrate its importance.
>
> **Response to W1(a):** We thank the reviewer for the helpful suggestions and apologize for the lack of clarity. This paper argues that previous studies on item-side group fairness (IGF) have **lacked consideration of user-side utility** that **depends on a user's social attributes.** The core intuition is that, **for a given item (in an item-group), recommending it to a user with "richer (more active)" social attributes will result in greater utility in practice, due to the fact that the user $u$'s interaction (e.g., clicks or purchases) may by affected by its neighbors $\mathcal{N}_u$.** This is also known as user conformity bias in previous debiased recommendation studies [Zheng et al. "Disentangling user interest and conformity for recommendation with causal embedding." WWW 2021].
>
> To tackle the above problem, the fairness metric studied in this paper considers both the exposure chance of the items and the social attributes of the corresponding recommended users, and is formalized as a multi-objective optimization problem. Taking the widely-studied statistical parity (SP) in IGF as an example, we not only want item-group A and item-group B to have the same exposure rates, but also want the users to whom item-group A is recommended to have similar social attributes as the users to whom item group B is recommended to. We hope the above clarification provides more understanding of the motivation for our work.
>
> > **W1(b):**
> While utilizing a multi-objective optimization method (e.g., Pareto optimization) is convincing, it is not novel enough. Similar methods have been widely used in recommender systems, like [1].
>
> **Response to W1(b):** The reviewer raises an interesting concern. We **agree** that using a multi-objective optimization (e.g., Pareto optimization) is not novel by itself. However, the novelties of this paper are:
> - (a) the proposed social-attribute aware IGF metrics (namely NSP and NEO) corresponding to SP and equal opportunity (EO), respectively, to take into account the direct and social attributes of the users;
> - (b) the proposed Pareto optimization, compared with [1], has more flexible and controllable (via pre-specified preference regions) trade-offs between SP and NSP (as well as between EO and NEO);
> - (c) theoretically demonstrates that the proposed optimization can achieve the Pareto optimal solution for SP and NSP (as well as for EO and NEO) while guaranteeing the least accuracy of the learned recommendation model.
>
> ### **Presentation**
>
> > **W2:** Some mathematical notations are not clear in this paper, making it hard to follow. For example, $G_{g_a}(i)$ in line 82 should reflect its relationship to $u$. Besides, $R_v$ in line 117 should reflect its relationship to $u$ and $i$.
>
> **Response to W2:** We thank the reviewers for the useful comments. **Yes, $R_v$ in line 117 should be $R_v(u, i)$ instead of $R_v(u)$.** However, **$G_{g_a}(i)$ in line 82 is the group indicator of item $i$,** i.e., equals to 1 if item $i$ belongs to item-group $g_a$, thus **should not** relies on $u$.
>
>
> ### **Experiment**
> > **W3(a):** Although BPRMF is a classical recommendation method, .... Exploring the fairness issue using popular deep-learning algorithms like NeuMF[2] and NGCF[3] may be more effective to evaluate the proposed method.
>
> > **Q1:** Why this paper chooses BRPMF to conduct your experiments rather than some novel deep-learning-based methods like NeuMF and NGCF?
>
> **Response to W3(a) and Q1:** Thanks for your helpful suggestion. **As suggested by the reviewer, we add extensive experiments by taking NeuMF and NGCF as backbones, respectively, to validate the effectiveness of our proposal.** Please kindly refer to the attachment PDF for more details.
>
> > **W3(b):** In the ablation study, when optimizing the model with only SP (i.e., w/o SP), the model achieves the best NSP fairness. The paper should analyze the reason for this.
>
> > **Q2:** Why optimizing the model with only SP can achieve the best NSP fairness?
>
> **Response to W3(b) and Q2:** We thank the reviewer for pointing out this issue. **We believe that there is a misunderstanding here due to our typo: "w/o" in this paper means "without", not "with only".** Therefore, the model that considers only NSP (i.e., w/o SP) achieves the optimal NSP fairness.
>
> > **W3(c):** The value of $R_v$ is important for the proposed method. The authors should conduct experiments with different values of this hyperparameter to prove why they set the default value to 1.
>
> > **Q3:** Why you didn’t conduct an experiment of the effect of the value of $R_v$?
>
> **Response to W3(c) and Q3:** We agree with you and **have added the experiments by set varying $R_v$ with different users, items, and user-item pairs, respectively.** Specifically, we set $R_v(u)$ equals number of items that user u has interacted with, $R_v(i)$ equals number of users that item i has interacted with, and $R_v(u, i)=R_v(u)*R_v(i)$, respectively. Please kindly refer to the attachment PDF for more details.
>
> ***
> **We hope the above discussion will fully address your concerns about our work, and we would really appreciate it if you could be generous in raising your score.** We look forward to your insightful and constructive responses to further help us improve the quality of our work. Thank you!

---

> > ### Author Response · Authors · 2023-08-18
> > **Supplementary experiments on varying $R_v$ choices**
> >
> > For ease of checking and discussion, we've opened a new reply about supplemental experiments.
> >
> > > The value of $R_v$ is important for the proposed method. The authors should conduct experiments with different values of this hyperparameter to prove why they set the default value to 1. Why you didn’t conduct an experiment of the effect of the value of
> >  $R_v$?
> >
> > -  **Response:** We thank the reviewer for pointing out this issue. **We have added the experiments by set varying $R_v$ with different users, items, and user-item pairs, respectively.** Specifically, we set $R_v(u)$ equals number of items that user $u$ has interacted with, $R_v(i)$ equals number of users that item $i$ has interacted with, and $R_v(u, i)=R_v(u) * R_v(i)$, respectively.
> >
> > |SP&NSP,Nv(u)|NDCG|SP|NSP|F1SP|deg|EO&NEO,Nv(u)|NDCG|EO|NEO|F1EO|deg|
> > |--|--|--|--|--|--|--|--|--|--|--|--|
> > |BPRMF|$\underline{0.2426}$|0.0966|0.1146|0.1049|49.9|BPRMF|**0.2426**|0.0189|0.1462|0.0334|82.7|
> > |+SPReg|0.2268|0.0070|0.0213|0.0105|71.8|+EOReg|0.2362|0.0026|0.1556|0.0051|89.0|
> > |+NSPReg|0.2318|0.0610|0.0448|0.0517|36.3|+NEOReg|0.2329|0.0084|0.0974|0.0155|85.0|
> > |+SP&NSPReg|0.2407|0.0058|0.0177|0.0088|71.7|+EO&NEOReg|0.2347|$\underline{0.0026}$|0.1232|$\underline{0.0051}$|88.8|
> > |+SPPost|0.2415|0.0075|0.0171|0.0104|66.4|+EOPost|$\underline{0.2386}$|0.0035|0.1541|0.0068|88.7|
> > |+NSPPost|0.2414|0.0736|0.0614|0.0669|39.9|+NEOPost|0.2385|0.0101|0.1171|0.0185|85.1|
> > |+SP&NSPPost|0.2418|0.0060|0.0204|0.0093|73.5|+EO&NEOPost|0.2369|0.0030|0.1141|0.0059|88.5|
> > |MOOMTL|0.2157|0.0071|0.0177|0.0101|68.3|MOOMTL|0.2353|0.0029|0.1542|0.0057|88.9|
> > |Ours(region0)|0.1943|0.1819|0.0185|0.0336|5.8|Ours(region0)|0.1421|0.1802|**0.0818**|0.1125|24.4|
> > |Ours(region1)|0.2363|0.0426|0.0305|0.0356|35.6|Ours(region1)|0.2055|0.1154|$\underline{0.0826}$|0.0963|35.6|
> > |Ours(region2)|0.2351|0.0226|0.0214|0.0220|43.4|Ours(region2)|0.2238|0.0831|0.1018|0.0915|50.8|
> > |Ours(region3)|0.2337|$\underline{0.0058}$|**0.0152**|**0.0084**|69.2|Ours(region3)|0.2287|0.0723|0.1406|0.0955|62.8|
> > |Ours(region4)|**0.2428**|**0.0057**|$\underline{0.0158}$|$\underline{0.0084}$|70.1|Ours(region4)|0.2357|**0.0025**|0.1483|**0.0050**|89.0|
> >
> >
> >
> > |SP&NSP,Nv(i)|NDCG|SP|NSP|F1SP|deg|EO&NEO,Nv(i)|NDCG|EO|NEO|F1EO|deg|
> > |--|--|--|--|--|--|--|--|--|--|--|--|
> > |BPRMF|$\underline{0.2426}$|0.0966|0.2992|0.1461|72.1|BPRMF|**0.2426**|0.0189|0.1242|0.0328|81.4|
> > |+SPReg|0.2268|0.0070|0.2423|0.0136|88.3|+EOReg|0.2362|$\underline{0.0026}$|0.1227|0.0051|88.8|
> > |+NSPReg|0.2213|0.0459|$\underline{0.2008}$|0.0748|77.1|+NEOReg|0.2320|0.0084|$\underline{0.1035}$|0.0155|85.4|
> > |+SP&NSPReg|0.2424|0.0063|0.2626|0.0123|88.6|+EO&NEOReg|0.2379|0.0036|0.1040|0.0070|88.0|
> > |+SPPost|0.2410|0.0055|0.2483|0.0108|88.7|+EOPost|$\underline{0.2386}$|0.0035|0.1216|0.0068|88.4|
> > |+NSPPost|0.2294|0.0622|0.2030|0.0952|73.0|+NEOPost|0.2292|0.0075|0.1138|0.0141|86.2|
> > |+SP&NSPPost|0.2402|0.0046|0.2501|0.0090|88.9|+EO&NEOPost|0.2369|0.0032|0.1097|0.0063|88.3|
> > |MOOMTL|0.2026|$\underline{0.0038}$|0.2411|$\underline{0.0076}$|89.1|MOOMTL|0.2348|0.0026|0.1114|$\underline{0.0051}$|88.7|
> > |Ours(region0)|0.1494|0.5332|**0.1681**|0.2556|17.5|Ours(region0)|0.0607|0.3022|0.1530|0.2031|26.9|
> > |Ours(region1)|0.2338|0.4381|0.3083|0.3619|35.1|Ours(region1)|0.2123|0.1676|**0.0986**|0.1242|30.5|
> > |Ours(region2)|0.2386|0.2435|0.3278|0.2794|53.4|Ours(region2)|0.2266|0.1203|0.1113|0.1156|42.8|
> > |Ours(region3)|**0.2428**|0.1067|0.3013|0.1576|70.5|Ours(region3)|0.2333|0.0648|0.1154|0.0830|60.7|
> > |Ours(region4)|0.2412|**0.0037**|0.2617|**0.0073**|89.2|Ours(region4)|0.2355|**0.0025**|0.1145|**0.0049**|88.7|
> >
> >
> >
> > |SP&NSP,Nv(u,i)|NDCG|SP|NSP|F1SP|deg|EO&NEO,Nv(u,i)|NDCG|EO|NEO|F1EO|deg|
> > |--|--|--|--|--|--|--|--|--|--|--|--|
> > |BPRMF|$\underline{0.2426}$|0.0966|0.2805|0.1437|71.0|BPRMF|**0.2426**|0.0189|0.1485|0.0335|82.8|
> > |+SPReg|0.2425|0.0059|0.2405|0.0115|88.6|+EOReg|0.2362|$\underline{0.0026}$|0.1510|$\underline{0.0051}$|89.0|
> > |+NSPReg|0.2380|0.0700|0.2119|0.1052|71.7|+NEOReg|0.2265|0.0062|0.0810|0.0116|85.6|
> > |+SP&NSPReg|0.2425|0.0068|0.2386|0.0132|88.4|+EO&NEOReg|0.2381|0.0039|0.0878|0.0074|87.5|
> > |+SPPost|0.2419|0.0049|0.2320|0.0096|88.8|+EOPost|$\underline{0.2386}$|0.0035|0.1496|0.0068|88.7|
> > |+NSPPost|0.2407|0.0853|0.2212|0.1231|68.9|+NEOPost|0.2310|0.0071|0.0909|0.0131|85.5|
> > |+SP&NSPPost|0.2404|0.0049|0.2399|0.0095|88.8|+EO&NEOPost|0.2367|0.0028|0.0863|0.0053|88.2|
> > |MOOMTL|0.2025|$\underline{0.0040}$|$\underline{0.2095}$|$\underline{0.0078}$|88.9|MOOMTL|0.2353|0.0029|0.1485|0.0057|88.9|
> > |Ours(region0)|0.1446|0.6141|**0.1932**|0.2939|17.5|Ours(region0)|0.1267|0.2184|0.0766|0.1134|19.3|
> > |Ours(region1)|0.2315|0.4201|0.2961|0.3474|35.2|Ours(region1)|0.2036|0.1472|**0.0526**|0.0775|19.7|
> > |Ours(region2)|0.2370|0.2043|0.2741|0.2341|53.3|Ours(region2)|0.1972|0.0927|0.1039|0.0980|48.3|
> > |Ours(region3)|**0.2426**|0.0909|0.2745|0.1366|71.7|Ours(region3)|0.2318|0.0467|$\underline{0.0714}$|0.0565|56.8|
> > |Ours(region4)|0.2408|**0.0038**|0.2399|**0.0074**|89.1|Ours(region4)|0.2356|**0.0025**|0.1395|**0.0050**|89.0|

---

> > > ### Author Response · Authors · 2023-08-18
> > > **Supplementary experiments on taking NeuMF[2] and NGCF[3] as backbones**
> > >
> > > For ease of checking and discussion, we've opened a new reply about supplemental experiments.
> > >
> > > > Why optimizing the model with only SP can achieve the best NSP fairness?
> > >
> > > **Response:** We apologize for making a **typo** here: **"w/o" in this paper means "without", not "with only".** Therefore, the model that considers only NSP (i.e., w/o SP) achieves the optimal NSP fairness.
> > >
> > > > Although BPRMF is a classical recommendation method, it is an outdated recommendation algorithm compared to deep-learning-based methods used in modern recommender systems. Therefore, mitigating fairness in BRPMF may have limited impact in today's systems. Exploring the fairness issue using popular deep-learning algorithms like NeuMF[2] and NGCF[3] may be more effective to evaluate the proposed method.
> > >
> > > [2] He X, Liao L, Zhang H, et al. Neural collaborative filtering[C]//Proceedings of the 26th international conference on world wide web. 2017: 173-182.
> > >
> > > [3] Wang X, He X, Wang M, et al. Neural graph collaborative filtering[C]//Proceedings of the 42nd international ACM SIGIR conference on Research and development in Information Retrieval. 2019: 165-174.
> > >
> > > **Response:** We thank the reviewer for pointing out this issue. **We have added extensive experiments by taking NeuMF[2] and NGCF[3] as backbones, respectively, to validate the effectiveness of our proposal.**
> > >
> > > |SP&NSP|NDCG|SP|NSP|F1SP|deg|EO&NEO|NDCG|EO|NEO|F1EO|deg|
> > > |---------------|-------------|-------------|-------------|-------------|----|---------------|-------------|-------------|-------------|-------------|----|
> > > |**NeuMF[2]**|0.2220|0.2013|0.2464|0.2216|50.8|**NeuMF**|**0.2220**|0.0291|0.1006|0.0451|73.9|
> > > |+SPReg|$\underline{0.2260}$|0.0157|0.0473|0.0235|71.7|+EOReg|0.2125|0.0034|0.1077|0.0066|88.2|
> > > |+NSPReg|0.2223|0.0229|0.0228|0.0228|44.9|+NEOReg|0.2191|0.0281|$\underline{0.0363}$|0.0317|52.2|
> > > |+SP&NSPReg|0.2256|0.0165|0.0620|0.0261|75.1|+EO&NEOReg|0.2191|0.0053|0.1034|0.0100|87.1|
> > > |+SPPost|0.2146|0.0151|0.0631|0.0243|76.6|+EOPost|0.2183|0.0042|0.1073|0.0082|87.7|
> > > |+NSPPost|0.2138|0.0686|0.0421|0.0522|31.5|+NEOPost|0.2142|0.0185|0.0384|0.0250|64.3|
> > > |+SP&NSPPost|0.2161|0.0187|0.0433|0.0262|66.6|+EO&NEOPost|0.2168|0.0044|0.0721|0.0084|86.5|
> > > |MOOMTL|0.1695|0.0177|0.0611|0.0274|73.9|MOOMTL|0.2113|$\underline{0.0034}$|0.1065|$\underline{0.0065}$|88.2|
> > > |Ours(region0)|0.1529|0.1064|**0.0166**|0.0287|8.9|Ours(region0)|0.1015|0.1130|**0.0361**|0.0547|17.7|
> > > |Ours(region1)|0.2237|0.0305|$\underline{0.0171}$|0.0219|29.2|Ours(region1)|0.1967|0.0830|0.0471|0.0601|29.6|
> > > |Ours(region2)|0.2211|0.0540|0.0659|0.0594|50.7|Ours(region2)|0.1917|0.1228|0.0926|0.1056|37.0|
> > > |Ours(region3)|0.2136|**0.0106**|0.0281|**0.0153**|69.4|Ours(region3)|0.2003|0.0744|0.1136|0.0899|56.8|
> > > |Ours(region4)|**0.2263**|$\underline{0.0139}$|0.0484|$\underline{0.0216}$|73.9|Ours(region4)|$\underline{0.2201}$|**0.0017**|0.0941|**0.0033**|89.0|
> > >
> > >
> > >
> > > |SP&NSP|NDCG|SP|NSP|F1SP|deg|EO&NEO|NDCG|EO|NEO|F1EO|deg|
> > > |---------------|-------------|-------------|-------------|-------------|----|---------------|-------------|-------------|-------------|-------------|----|
> > > |**NGCF[3]**|0.2389|0.1164|0.1356|0.1253|49.4|**NGCF**|**0.2389**|0.0210|0.1045|0.0349|78.7|
> > > |+SPReg|0.2372|0.0034|0.0202|0.0058|80.5|+EOReg|0.2351|0.0019|0.1110|0.0037|89.0|
> > > |+NSPReg|0.2318|0.0109|0.0114|0.0111|46.3|+NEOReg|0.2329|0.0092|0.0666|0.0161|82.2|
> > > |+SP&NSPReg|0.2392|0.0044|0.0185|0.0072|76.5|+EO&NEOReg|0.2338|0.0022|0.0763|0.0043|88.4|
> > > |+SPPost|0.2378|0.0048|0.0253|0.0081|79.3|+EOPost|$\underline{0.2352}$|0.0027|0.1118|0.0054|88.6|
> > > |+NSPPost|0.2353|0.0641|0.0369|0.0468|29.9|+NEOPost|0.2349|0.0105|0.0706|0.0182|81.6|
> > > |+SP&NSPPost|0.2376|0.0226|0.0221|0.0224|44.4|+EO&NEOPost|0.2333|0.0021|0.0766|0.0041|88.4|
> > > |MOOMTL|0.2256|$\underline{0.0032}$|$\underline{0.0092}$|$\underline{0.0048}$|70.8|MOOMTL|0.2106|$\underline{0.0015}$|0.1060|$\underline{0.0029}$|89.2|
> > > |Ours(region0)|0.2135|0.0680|0.0201|0.0310|16.4|Ours(region0)|0.1090|0.1782|0.1030|0.1305|30.0|
> > > |Ours(region1)|0.2383|0.0214|0.0128|0.0160|30.9|Ours(region1)|0.2065|0.0663|**0.0338**|0.0448|27.0|
> > > |Ours(region2)|**0.2405**|0.0134|0.0120|0.0126|41.9|Ours(region2)|0.1871|0.0353|$\underline{0.0457}$|0.0398|52.3|
> > > |Ours(region3)|$\underline{0.2397}$|0.0152|0.0294|0.0201|62.6|Ours(region3)|0.2200|0.0443|0.0694|0.0541|57.5|
> > > |Ours(region4)|0.2155|**0.0019**|**0.0081**|**0.0031**|76.6|Ours(region4)|0.2347|**0.0013**|0.1123|**0.0026**|89.3|
> > >
> > > ***
> > >
> > > **We hope the above clarification will fully address your concerns about our work, and we would really appreciate it if you could thus upgrading your score. We are more than willing to your response confirming whether the above addresses your concerns. Thank you!**

---

### Official Review · Reviewer_pts6 · 2023-07-08

**Soundness:** 3 good
**Presentation:** 2 fair
**Contribution:** 2 fair
**Rating:** 2
**Confidence:** 5

**Summary:**

This paper introduces a new problem of fairness in recommendation tasks, specifically considering the social attribute of users and its influence on item exposures. The paper proposes social attribute-aware fairness metrics and formulates a multi-objective optimization problem to train recommender models with customizable trade-offs between direct and social utilities. The paper presents a new optimization algorithm and provides theoretical guarantees. Experimental results on real-world datasets demonstrate the effectiveness of the proposed approach.

**Strengths:**

(1) The paper addresses an important and timely topic of fairness in recommendation systems, taking into account the social attribute of users.
(2) The introduction of social attribute-aware fairness metrics provides a more comprehensive and holistic view of fairness in recommendation tasks.
(3) The formulation of a multi-objective optimization problem allows for flexible customization of the trade-off between direct and social utilities, providing a practical solution for real-world scenarios.

**Weaknesses:**

(1) The paper lacks clarity in the presentation of the optimization algorithm and its theoretical guarantee. More detailed explanations and proofs would be helpful.
(2) The evaluation section could be improved with a more comprehensive comparison with existing baselines in the field of fairness-aware recommendation systems.
(3) The implementation details are not enough.
(4) The writing should be improved.
(5) Please discuss more about the application scenario of the proposed method.
(6) Please use more datasets to provide more solid results.

**Questions:**

My concerns involves clarity, expeirments, implementations, etc.. Please refer to the above comments.

---

> ### Author Rebuttal · Authors · 2023-08-10
>
> We sincerely thank you for the helpful suggestions. **As suggested by the reviewer, we have added extensive discussions and experiments to address all your concerns.** Below, we hope to address your concerns and questions to improve the clarity and quality of our paper.
>
> ### **Methodology**
>
> > **W1:** The paper lacks clarity in the presentation of the optimization algorithm and its theoretical guarantee. More detailed explanations and proofs would be helpful.
>
> **Response**: Setting other optimization objectives makes it difficult to theoretically achieve Pareto optimality, and the proposed method has strict theoretical guarantees. Please kindly refer to lines 185-190 in our manuscript.
>
> ### **Novelty**
>
> **Response**: We kindly remind the reviewer that the novelties of this paper are:
> - (a) the proposed social-attribute aware IGF metrics (namely NSP and NEO) corresponding to SP and equal opportunity (EO), respectively, to take into account the direct and social attributes of the users;
> - (b) the proposed Pareto optimization, compared with [1], has more flexible and controllable (via pre-specified preference regions) trade-offs between SP and NSP (as well as between EO and NEO);
> - (c) theoretically demonstrates that the proposed optimization can achieve the Pareto optimal solution for SP and NSP (as well as for EO and NEO) while guaranteeing the least accuracy of the learned recommendation model.
>
> ### **Experiments**
> > **W2:** The evaluation section could be improved with a more comprehensive comparison with existing baselines in the field of fairness-aware recommendation systems.
>
> > **W3:** The implementation details are not enough.
>
> > **W5:** Please discuss more about the application scenario of the proposed method.
>
> > **W6:** Please use more datasets to provide more solid results.
>
> **Response**: Thanks for your comments.
>
> -	Due to the page limits, we **postpone the detailed dataset descriptions in Appendix B1 in supplementary material,** including their origination, processing procedures, statistics, grouping method and social attributes.
>
> -	**As suggested by the reviewer, we add extensive experiments by taking NeuMF and NGCF as backbone model, respectively.** Moreover, we **set varying $R_v$ with different users, items, and user-item pairs, respectively, to validate the effectiveness of our method.** Please kindly refer to the attachment PDF for more details. From the tables in the attached pdf, we can observe that our method can always achieve the best fairness performance (i.e. minimum SP/NSP/F1SP/EO/NEO/F1EO values) and diverse trade-off between two sets of IGF metrics compared to other baselines. Additionally, in EO&NEO cases, our method may not obtain the best NDCG value apart from the basemodel, and we attribute this to the fairness-utility trade-off phenemenon; despite from that, our method can always get optimal NDCG values in SP&NSP settings.
>
> -	More discussion on Fig. 2: the figures in Fig. 2 is divided into four regions by two reference lines, which represents the corresponding metric value of x/y-axis obtained by single-objective optimization methods. More specifically, in the 1st and 3rd subfigure, the hierarchical line refers to the -log2NSP value of NSP Reg baseline, and the vertical line refers to the -log2SP value of SP Reg baseline; in the 2nd and 4th subfigure, we take the NDCG and -log2F1SP value of SP Reg baseline for reference. As for the divided four regions, the solutions in upper right (lower left) region can achieve better (worse) performance than both two reference lines (i.e. Pareto dominance), and the solutions in upper right / upper left / lower right region are regarded as Pareto optimal solutions.
>
> ### **Presentation**
> > **W4:** The writing should be improved.
>
> **Response**: Thank you for your comments. We apologize for the lack of clarity. We have made careful corrections throughout our manuscript.
>
> ***
>
> **We hope the above discussion will fully address your concerns about our work, and we would really appreciate it if you could be generous in raising your score.** We look forward to your insightful and constructive responses to further help us improve the quality of our work. Thank you!

---

> > ### Comment · Reviewer_pts6 · 2023-08-22
> > **Response to the Author Rebuttal**
> >
> > Dear Authors,
> >
> > Thanks for the reply. Your reply has partly addressed some of my concerns.
> > I have also noticed that Reviewer cJkK has the similar concerns with me, including idea, experiment, etc. I agree with him/her, and thus I keep my score unchanged.

---

### Author Rebuttal · Authors · 2023-08-10

We sincerely thank all the reviewers for valuable comments and suggestions! Below we have tried our best to address the concerns point-to-point for each reviewer, and we are looking forward for further discussion about our work. In addition, we have uploaded a page of supplementary material here, which contains some additional experimental results as suggested by reviewer cJkK and PYwN.

---

### Decision · Program_Chairs · 2023-09-21

**Decision:**

Accept (poster)

**Comment:**

The paper proposes new social attribute-aware item-side group fairness (IGF) metrics to account for influence of users' social attributes on item exposure utility. The paper extends existing item side fairness notions in recommender systems to incorporate social utility that is derived from a user's neighbors in the social network. In the scenarios where social utility impacts exposure of items indirectly (as compared to recommendations themselves), the existing notions of fairness see limitations and the proposed methods claims to fix that. The reviewers appreciated that the paper validates the proposed metrics and approach with extensive experiments on real datasets that show effectiveness of the proposed flexible and controllable approach. However, reviewers raised some concerns. There were some concerns about the clarity of the paper that the authors are advised to carefully consider for a final version. It might help readability if non-standard acronyms (MOOMTL, NSP Reg, NSP Post, etc.) are used sparingly since they are harder to keep track of throughout the paper.
Thank you for addressing the questions asked by the reviewers in their initial reviews. Please make changes to the paper and appendices accordingly, including adding the new experimental results that were shared in the discussion.